# Drainage of inflammatory macromolecules from the brain to periphery targets the liver for macrophage infiltration

Linlin Yang[1], Jessica A Jiménez[1], Alison M Earley[1], Victoria Hamlin[1], Victoria Kwon[1], Cameron T Dixon[1], Celia E Shiau[1,2]*

[1]Department of Biology, University of North Carolina at Chapel Hill, Chapel Hill, United States; [2]Department of Microbiology and Immunology, University of North Carolina at Chapel Hill, Chapel Hill, United States

**Abstract** Many brain pathologies are associated with liver damage, but a direct link has long remained elusive. Here, we establish a new paradigm for interrogating brain-periphery interactions by leveraging zebrafish for its unparalleled access to the intact whole animal for in vivo analysis in real time after triggering focal brain inflammation. Using traceable lipopolysaccharides (LPS), we reveal that drainage of these inflammatory macromolecules from the brain led to a strikingly robust peripheral infiltration of macrophages into the liver independent of Kupffer cells. We further demonstrate that this macrophage recruitment requires signaling from the cytokine IL-34 and Toll-like receptor adaptor MyD88, and occurs in coordination with neutrophils. These results highlight the possibility for circulation of brain-derived substances to serve as a rapid mode of communication from brain to the liver. Understanding how the brain engages the periphery at times of danger may offer new perspectives for detecting and treating brain pathologies.

*For correspondence: shiauce@unc.edu

Competing interests: The authors declare that no competing interests exist.

## Introduction

Whether a diseased or injured brain transmits signals to the periphery to activate a response is an interesting prospect in understanding brain-periphery communication, but remains underexplored. Interestingly, liver damage and neutrophil recruitment to the liver are common features in sepsis-related injury (*McDonald et al., 2012*) and several central nervous system (CNS) pathologies (*Campbell et al., 2010*; *Estrada et al., 2019*), including traumatic brain injury, multiple sclerosis, and Alzheimer's disease. A few recent studies have implicated a systemic, albeit most prominently a hepatic response to CNS inflammation due to CNS trauma or injury in mammals (*Campbell et al., 2005*; *Anthony et al., 2012*). However, the underlying mechanisms that may link brain inflammation with liver impairment remain unclear. The ability to track the cellular and molecular processes from the brain to the liver in vivo provides a direct means to understand the brain-liver association.

To investigate components of communication between the brain and periphery, we hypothesized that macrophages, key innate immune cells, capable of long-range migration and signaling (*Eom and Parichy, 2017*), could act as mediators of brain-periphery communication. To this end, we investigated if a brain perturbation triggering inflammation, using a brain-localized LPS microinjection as an experimental means, could trigger a peripheral organ response mediated by macrophages. We employed the zebrafish because it offers unparalleled access to in vivo tracking and manipulation of molecular and cellular processes in the intact whole vertebrate animal from brain to peripheral organs, which are largely conserved from zebrafish to human (*Santoriello and*

*Zon, 2012*). Using zebrafish, we were able to directly capture the dynamic changes in macrophages occurring in the body after a focal brain challenge and found the liver to be the most prominent target peripheral organ for immune infiltration. Our data support the notion that infiltrating and resident macrophages in the liver may critically modulate CNS and systemic inflammation by shaping the hepatic response to circulating or widely distributed molecules that may be infectious, toxic, or exogenous.

A possible route through which the brain may affect liver function may simply be a drainage of effector molecules into circulation, albeit even at a trace level, to initiate systemic inflammation, rather than direct brain to liver signaling. While much research has focused on mechanisms penetrating the blood-brain barrier (BBB) or the blood-cerebrospinal fluid (BCSF) barrier to enable entry of a peripheral agent into the brain parenchyma such as those relating to infectious diseases causing brain dysfunction (for example, by neurotropic viruses including *Rabies lyssavirus*, West Nile virus, and cytomegalovirus) and drug delivery to the brain (*Hladky and Barrand, 2016*; *van den Pol, 2009*), far less attention has been given to investigating the reciprocal transfer from brain to circulation and its consequences. Limiting the free passage of solutes and large molecules into the CNS is tightly-regulated by both BBB and BCSF barriers to ensure protection of the CNS from inappropriate tissue damage and inflammation. By contrast, the removal of waste and toxic agents from the brain interstitial space to ensure normal brain health and function requires appropriate metabolism or efflux of these substances (*Hladky and Barrand, 2018*). Previous studies on movement of CNS fluid and solutes indicate several possible routes of drainage of substances from the brain parenchyma including perivascular pathways that may exit through the CSF or lymph tracts, and the BBB (*Hladky and Barrand, 2018*). Therefore, if the net outcome is some degree of systemic inflammation due to an efflux of inflammatory cues from the brain to circulation, even at a low level, then it could effectively target the liver, but this remains to be investigated.

Given that the liver is equipped to process a large fraction of the total blood circulation from two major blood supplies (*Eipel et al., 2010*), the hepatic artery and portal vein from the gastrointestinal tract, it may not be surprising that the liver would be highly sensitive and responsive to inflammatory mediators and foreign agents in the blood flow. In fact, common and infectious bacteria (including *Escherichia coli*, *Klebsiella pneumoniae*, *Salmonella typhimurium* and *Listeria monocytogenes*) in the bloodstream have often been found to be cleared by the liver (*Adams, 2003*; *Gregory et al., 1996*). However, the mechanisms that specifically make the liver susceptible to systemic inflammation remain incompletely understood. Besides metabolic functions, the liver provides critical immune surveillance by recognizing and clearing away infectious, toxic, and microbial substances in the blood, a function that has largely been attributed to the Kupffer cells (*Bilzer et al., 2006*; *Kubes and Jenne, 2018*; *Racanelli and Rehermann, 2006*). Systemic inflammation stemming from infection, toxic insults, and autoimmunity (*Edwards and Wanless, 2013*) can cause chronic infiltration of leukocytes into the liver leading to liver damage and subsequent progression to fibrosis, cirrhosis or liver cancer (*Karlmark et al., 2008*; *Mossanen and Tacke, 2013*; *Huang et al., 2016*). However, how the liver responds and contributes to systemic inflammation by way of leukocyte infiltration remains poorly understood, a process that has not been directly visualized in vivo for an open dissection.

Here, we reveal new insights into the cellular dynamics and critical roles of the IL-34 and MyD88 signaling pathways as well as Kupffer-cell-independent mechanisms in mediating immune infiltration of the liver in response to systemic lipopolysaccharides (LPS), classic pro-inflammatory bacteria-derived stimuli, after brain intraparenchymal LPS microinjection. By using fluorescently traceable LPS in the brain as an experimental paradigm, we show that inflammatory cues could originate from the brain and trigger immune infiltration of the liver, a process not previously appreciated involving drainage of the macromolecules from brain to circulation. Using comparative analyses, time-lapse imaging, and blocking circulation, we found that the effects of brain-LPS injection were largely recapitulated by intravenous LPS injection, and stemmed from systemic inflammation akin to sepsis or endotoxemia. We can block infiltration of immune cells into the liver by disrupting MyD88, a key adaptor of Toll-like receptors responsible for LPS recognition, or by eliminating the cytokine IL-34 pathway, and by reducing inflammation pharmacologically. Additionally, coordination between macrophages and neutrophils is also essential for the liver infiltration. Infiltration by macrophages and neutrophils negatively impacts the liver by promoting inflammation and disrupting normal hepatic growth. Taken together, changes in macrophage behavior involving immune cell infiltration of the

liver may be triggered by drainage of inflammatory cues coming from the brain, providing a possible readout for an altered brain.

## Results

### Brain immune activation is associated with macrophages infiltrating the liver prior to Kupffer cell establishment

Reciprocal connections between brain and liver are apparent in various conditions, including encephalopathy and encephalitis after severe liver damage (*Butterworth, 2013*; *Felipo, 2013*), liver disruption after traumatic brain injury (*Villapol, 2016*; *Lustenberger et al., 2011*), and intracerebral injection of pro-inflammatory cytokines (*Campbell et al., 2005*). However, the routes of communication directly linking brain to liver remain poorly understood. To investigate one possible avenue of this, we sought to determine whether a brain perturbation such as inflammation could trigger macrophage activities corresponding to a response by the liver or other peripheral organs. To this end, we directly microinjected bacterial lipopolysaccharides (LPS) or *E. Coli* cells, well-established immune activators, into the brain tectum at four days post-fertilization (dpf) at a stage known to have a well formed brain with an established BBB (*Jeong et al., 2008*) and choroid plexus/ventricle system (*Henson et al., 2014*; *Fame et al., 2016*) as previously described (*Earley et al., 2018*). Stage-matched animals that were not injected or injected with the water vehicle in the brain were used as the control group (*Figure 1* and *Figure 1—figure supplement 1*). To analyze a possible change in peripheral macrophages after brain-LPS injection at four dpf, we used whole-mount RNA in situ hybridization for a macrophage marker, *mfap4*, to characterize the macrophage distribution in the whole body. Strikingly, we found a robust and distinctive aggregation of macrophages in the liver and near the brain injection site, but not apparently elsewhere (*Figure 1—figure supplement 1*). By contrast, the uninjected and water injected controls at four dpf were devoid of macrophages in the liver (*Figure 1—figure supplement 1*). We also found at least some of these macrophages in the liver to be activated based on their expression of the mitochondrial enzyme gene *irg1/acod1* (*Figure 1—figure supplement 1*) known to be highly upregulated during inflammation and specifically induced in inflammatory macrophages in zebrafish (*Lampropoulou et al., 2016*; *Sanderson et al., 2015*). Both injections of LPS or live *E. coli* cells into the brain led to macrophage presence in the liver, albeit LPS effects were consistently stronger (*Figure 1—figure supplement 1*).

To relate macrophage presence in the liver after brain-LPS injection to macrophages that normally reside in the liver (which we refer to as Kupffer cells), we analyzed the developmental timing of Kupffer cells, which had not been previously described, starting at the four dpf larval stage to the 40 dpf juvenile adult stage (*Figure 1a*). We found normally an absence of Kupffer cells at the stage of our brain microinjection at four dpf ($0.2 \pm 0.4$ standard deviation (s.d.) per liver), few to none at 5–6 dpf ($4.0 \pm 2.2$ s.d. per liver), and once larvae were fed and moved into the fish facility, Kupffer cell numbers grew substantially ($78.2 \pm 14.9$ s.d. per area of liver at 26 dpf) reaching to hundreds per liver in the juvenile adults (26–40 dpf at 9–12.5 mm standard length) (*Figure 1a*). Prior to this work, Kupffer cells were thought to be missing or sparse in zebrafish and other teleost species (*Goessling and Sadler, 2015*; *Sakano and Fujita, 1982*), but recent work tracing adult zebrafish Kupffer cells to their hematopoietic origin (*He et al., 2018*), and the data presented here collectively provide the first evidence for the prevalence of Kupffer cells in the zebrafish liver akin to their mammalian counterpart. Since the brain-LPS injection and subsequent analysis were conducted at four dpf, before the establishment of Kupffer cells, the presence of macrophages in the liver was likely because of active recruitment of peripheral macrophages and monocytes.

We subsequently conducted a time-course analysis of macrophage activities after brain-LPS injection at four dpf to determine if macrophages were actively infiltrating the liver. Using in vivo static and time-lapse imaging in double transgenic zebrafish expressing both the macrophage reporter *mpeg1:GFP* and the liver hepatocyte reporter *fabp10a:DsRed*, we observed macrophages actively migrating or circulating into the liver, affirming the in situ results (*Figure 1—figure supplement 1*). We captured macrophage dynamics in the liver region at 8–10 hr post-injection (hpi) of LPS in the brain (*Video 1* and *Figure 1—figure supplement 2*) when substantial numbers of infiltrating macrophages can be observed. Conversely, the brain-water injected controls mostly had zero macrophages in the liver (average of $1.4 \pm 1.2$ standard deviation (s.d.) macrophages per liver) compared

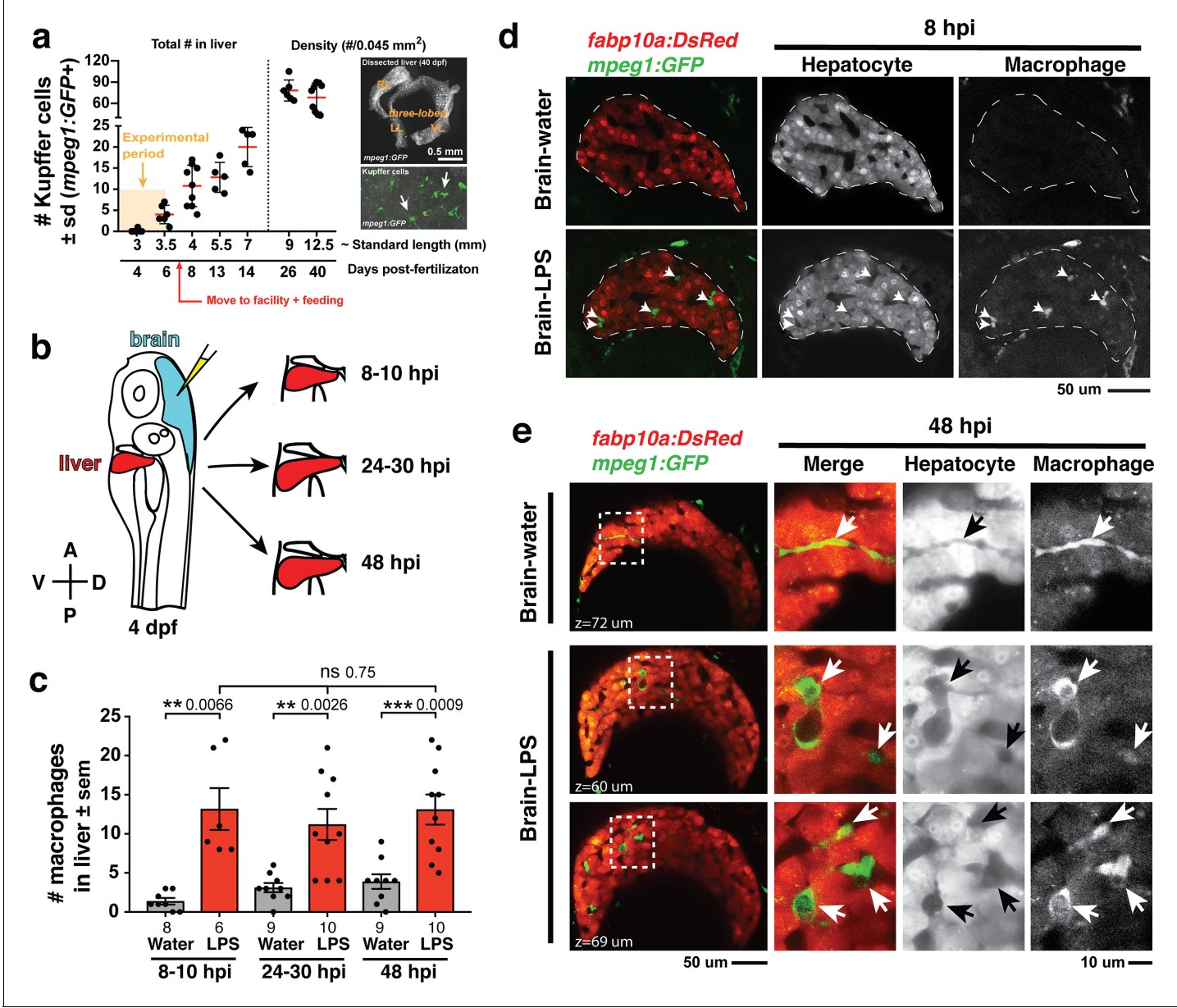

**Figure 1.** Induction of brain inflammation triggers macrophage infiltration into the liver. (a) Time-course of Kupffer cell development. Total macrophage numbers (*mpeg1:GFP*+) per liver from 4 to 14 days post-fertilization (dpf) and macrophage density (number per field of view) from dissected livers at juvenile adult stages from 26 to 40 dpf. Standard length corresponding to each stage is shown. Kupffer cells are not present at four dpf at the time of brain microinjection and during most of the experimental period (orange box). Feeding began after six dpf to ensure normal animal development. Fluorescent images on the right show dissected whole liver with the typical three-lobed structure at 40 dpf (top) and high magnification of top dotted box region showing Kupffer cells (bottom). LL, Left lobe; RL, right lobe; VL, ventral lobe. (b) Schematic of brain microinjection at four dpf and analysis of the hepatic response at 8–10 hr post injection (hpi), 24–30 hpi, and 48 hpi. A, anterior; P, posterior; V, ventral; D, dorsal. (c) Quantification of macrophage infiltration in the liver comparing between LPS and control water injections in the brain at four dpf and analyzed at different timepoints. Numbers below bar graphs represent *n*, number of animals analyzed. (d) At eight hpi, single-plane image from a z-stack shows infiltrated macrophages (GFP+, arrows) nested between hepatocytes (DsRed+) in the liver (dotted region) after brain-LPS injection, but no macrophages observed in control brain-water injection. (e) At 48 hpi, images from two separate z-planes show an abnormally large number of macrophages in the liver that persists after brain-LPS injection (arrows), while few presumably Kupffer cells begin to appear in control brain-water injected animals at this timepoint (arrow). Two-tailed Welch's t-test was used to determine statistical significance for each pair-wise comparison. One-way ANOVA test for comparing the three LPS injection groups. sem, standard error of means; ns, not significant; **, p<0.01; ***, p<0.001.

The online version of this article includes the following figure supplement(s) for figure 1:

*Figure 1 continued on next page*

*Figure 1 continued*

**Figure supplement 1.** Whole-body analysis using RNA in situ hybridization shows abnormal localization of macrophages in the liver after brain injection of LPS or bacteria.

**Figure supplement 2.** In vivo time-lapse imaging shows dynamic movements and processes of infiltrating macrophages in the liver at short- (10 hpi) and long- (48 hpi) term timepoints after brain-LPS injection.

**Figure supplement 3.** Brain-LPS injection was not found to induce *mfap4* expression in the liver.

**Figure supplement 4.** Macrophages infiltrate the liver through vasculature and vasculature-independent routes.

**Figure supplement 5.** Long-term in vivo tracking of infiltrating macrophages shows that occupation time in the liver can be used for their classification.

with an average of 13.2 ± 6.6 s.d. macrophages per liver after brain-LPS injection (*Figure 1b–e*). Later at 24–30 hpi and 48 hpi, significant numbers of macrophages in the liver persisted even two days after brain-LPS injection (11.2 ± 6.3 s.d. and 13.1 ± 6.1 s.d. macrophages per liver, respectively) (*Figure 1c*, *Video 2* and *Figure 1—figure supplement 2*). At these later timepoints, a few Kupffer cells may begin to emerge at less than five per liver (*Figure 1c*). In agreement with the timeline of Kupffer cell development (*Figure 1a*), only a few macrophages were detectable in uninjected and water-injected controls at the two later timepoints (3.1 ± 1.8 s.d. and 3.9 ± 2.8 s.d. macrophages per liver, respectively) (*Figure 1c*). Macrophages in the liver at 48 hpi appeared more stationary than earlier at 8–10 hpi (*Videos 1* and *2*, and *Figure 1—figure supplement 2*). In all timepoints of analysis, infiltrated macrophages in the liver were found in several locations, including inside the sinusoids (liver microvessels) similar to that described during mammalian liver injury (*Iwakiri et al., 2014*), and surprisingly also in the parenchyma intermingling with hepatocytes (*Figure 1d–e*, *Videos 1* and *2*, *Figure 1—figure supplement 2*), a macrophage behavior previously not known. Due to some examples of broad liver *mfap4* in situ expression (*Figure 1—figure supplement 1*) in comparison to a discrete number of infiltrating macrophages by live imaging after brain-LPS injection (*Figure 1*), we assessed whether this could be explained by an induction of ectopic *mfap4* expression in the liver upon LPS activation (*Figure 1—figure supplement 3*). Using a transgenic line *mfap4:tdTomato* to mark cells expressing the *mfap4* gene, we found *mfap4* restricted to macrophages and absent in liver cells (*Figure 1—figure supplement 3*), suggesting the broad liver *mfap4* expression may be diffuse in situ signals coming from a liver more densely populated by infiltrated macrophages. To analyze the nature of physical contact of infiltrated macrophages to the hepatic sinusoids after brain-LPS injection, we imaged double transgenic zebrafish expressing the endothelial (*kdrl:mCherry*) and macrophage (*mpeg1:GFP*) reporters. Time-lapse imaging showed that macrophages can actively infiltrate the hepatic sinusoids, as well as be associated or entirely independent of the hepatic vasculature (*Figure 1—figure supplement 4* and *Video 3*).

To determine whether this infiltration endured over time, we examined the number of liver-infiltrating macrophages over a two-day period after brain-LPS injection. Results indicate no significant change in macrophage presence in liver, suggesting that infiltrated macrophages either stayed in the liver, moved in and out of the liver at similar rates, or both. To distinguish these possibilities, we tracked these infiltrating macrophages directly in vivo over a continuous 10 hr period after brain-LPS injection at four dpf starting at 12 hpi. Using live cell tracking, we found a small number (~13%) of infiltrating macrophages that stayed in the liver longer than 2 hr, including occasional infiltrates which remained in the liver past the total duration of imaging (>9 hr) (*Figure 1—figure supplement 5*

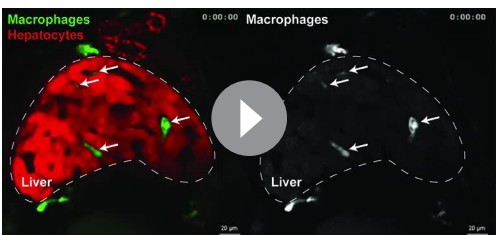

**Video 1.** Time-lapse imaging of macrophages infiltrating the liver 10 hr after brain-LPS injection. Representative single z-plane through the liver (*fabp10a:DsRed*+) shown from confocal imaging of one z-stack every 1 min and 15 s for ~1 hr using a 40x objective. A range of dynamic macrophage (*mpeg1:GFP*+) behaviors is shown: some nestled in gaps between hepatocytes presumably in the sinusoids while others either circulate or traverse the liver back and forth with long processes. Left panel shows the merge channel and the right panel shows single GFP channel for macrophages. Movie file shown at 30 fps. See *Figure 1—figure supplement 2* for additional description. Arrows, infiltrated macrophages. Dotted line, liver area.
https://elifesciences.org/articles/58191#video1

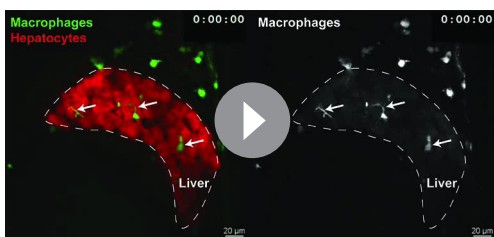

**Video 2.** Time-lapse imaging of macrophages infiltrating the liver 48 hr after brain-LPS injection. Representative single z-plane through the liver (*fabp10a:DsRed+*) shown from confocal imaging of one z-stack every 1 min for ~1 hr using a 40x objective. Macrophages (*mpeg1:GFP+*) in the liver (*fabp10a:DsRed+*) appear to be more stationary than at the earlier timepoint at eight hpi. Varied morphology still apparent from individual macrophages with long processes to a rounded cell shape with little to no apparent processes. Movie file shown at 30 fps. See *Figure 1—figure supplement 2* for additional description.
https://elifesciences.org/articles/58191#video2

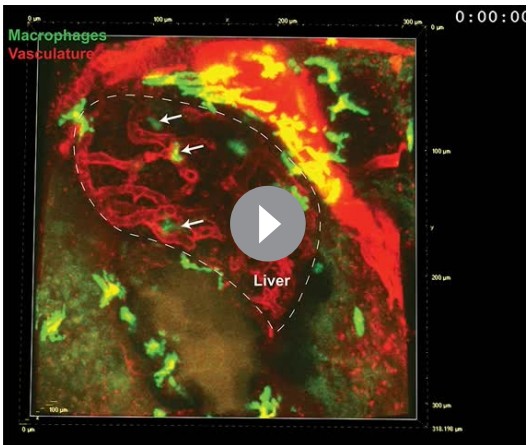

**Video 3.** Association of infiltrating macrophages with the vasculature in vivo 8 hr after brain-LPS injection. 3D view of a time-lapse imaging of an 80 μm volume from four dpf zebrafish injected with LPS showing the liver region encompassing the hepatic vasculature (*kdrl:mCherry+*) and macrophages (*mpeg1:GFP+*). One z-stack was acquired every 90 s for over a 5 hr period using a 40x objective. Three types of macrophage association with vasculature observed: inside, associated, or independent of vasculature. See *Figure 1—figure supplement 4* for additional description.
https://elifesciences.org/articles/58191#video3

and *Video 4*). Infiltrating macrophages were coined 'residing' when they occupied the liver for more than 2 hr, and these were on average moving slower at 1.1 ± 0.7 μm/min than the 'transient' population at 2.2 ± 1.1 μm/min which occupied the liver for more than one timepoint (2 min) but less than 2 hr (*Figure 1—figure supplement 5*). We found a significant fraction (~30%) of infiltrating macrophages to be of the 'transient' type, while the majority at 57.4% were circulating (detected only in a single timepoint) (*Figure 1—figure supplement 5*). Both the 'residing' and 'transient' populations were not truly stationary but rather moved dynamically back and forth across the liver parenchyma amounting to large total distances traveled over time (309.2 ± 285.9 μm s.d. and 62.8 ± 66.7 μm s.d. in 560 min of tracking, respectively) (*Figure 1—figure supplement 5* and *Video 4*). These results indicated that both the presence of short- and long-term occupying macrophages accounted for the sustained large macrophage number in the liver even two days after brain-LPS injection (*Figure 1c*).

## Microinjection of LPS into brain leads to systemic LPS distribution triggering immune infiltration of the liver

To understand how LPS in the brain may lead to liver effects, we first determined whether the injected LPS remained restricted to the brain or

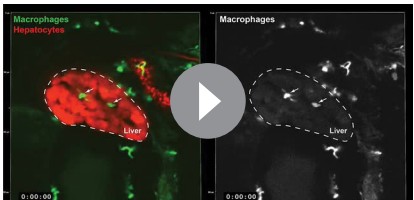

**Video 4.** In vivo long-term tracking of infiltrating macrophages in the liver at 12 hr after brain-LPS injection for a 10 hr continuous period. 3D view of a time-lapse imaging corresponding to a 6 μm volume collected every 2 min for a 10 hr period starting at 12 hpi at four dpf using a 40x objective. Left panel, merged channel for hepatocytes (*fabp10a:DsRed*) and macrophages (*mpeg1:GFP*). Right panel, GFP channel alone for showing macrophages. Dextran-Alexa 568 was co-injected into the brain as a tracer to validate injections, and can be seen labeling the pronephros in the DsRed channel. Circulating, transient, and residing macrophages can be observed within the liver tissue with varied cellular dynamics from being rounded and rapidly flowing through the liver to being ramified and migrating back and forth traversing the liver, respectively. See *Figure 1—figure supplement 5* for additional description. Movie file shown at 30 fps.
https://elifesciences.org/articles/58191#video4

possibly transferred to the periphery over time. We used fluorescently tagged LPS to directly track the LPS molecules after brain tectum microinjection in the whole body for a continuous 24 hr period (*Figure 2* and *Video 5*). This provides a means to visualize binding, transport, and internalization of LPS in the brain and body. Fluorescently tagged dextran was used as a control tracer to analyze the general molecular distribution independent of LPS (*Figure 2* and *Video 6*) and to verify successful injections. Initially the LPS macromolecules injected into the brain parenchyma were restricted to the focal location of the injection site but quickly within seconds they filled the cerebral ventricles joining the cerebrospinal fluid (CSF) as they continue to flow into the spinal canal in an anteroposterior direction (*Figure 2* and *Video 7*), thus our zebrafish brain microinjections are comparable to mammalian intracerebroventricular injections (*Glascock et al., 2011*).

To evaluate the dynamics and major routes of LPS passage to the general circulation, we used high-speed and high-sensitivity stereomicroscopy to trace the movement of fluorescently tagged LPS in real time at one frame per second starting before the brain tectum microinjection to almost one hour after the injection (*Figure 2c–d* and *Video 7*). While most of the LPS remained restricted within the ventricular system, we found the hindbrain ventricle (hbv) to be the key region from which LPS spread into the parenchyma and surrounding interstitial space, especially in the dorsal-most portion of the junction between the hindbrain and spinal cord encompassing the dorsal longitudinal anastomotic vessels (DLAVs) and the dorsal longitudinal lymphatic vessel (DLLV) (*Figure 2—figure supplements 1* and *2*, and *Video 7*). LPS at a low level were detected to exude out from the hbv into the hindbrain interstitial fluid (ISF) starting at around 1 min after injection (*Video 7* and *Figure 2c*) and reached a maximum of about 30% of LPS injection level at 50 min after injection (*Figure 2—figure supplement 3*). After a delay of about 13 min after injection, the first LPS fluorescence signals were faintly measured in the pronephros, a region assessed as a proxy for general circulation (*Figure 2c–d* and *Video 7*). Peripheral LPS level increased over time but remained low at less than 15% of the LPS injection level (*Figure 2—figure supplement 3*).

To characterize LPS at the tissue and cellular level along its outflow path, we used high power confocal imaging on the same LPS-injected zebrafish which were tracked by stereomicroscopy (*Figure 2* and *Video 7*) to carry out further imaging at later timepoints. Interestingly, LPS molecules were taken up by brain lymphatic endothelial cells (blec), also known as fluorescent granular perithelial cells (FGPs), as well as by facial and trunk lymphatic vessels as shown by co-localization of LPS with the lymphatic reporter *mrc1a:GFP* at 1.5 hpi (*Jung et al., 2017*; *Figure 2—figure supplements 1* and *2*), suggesting LPS exited through these lymphatic structures. These results were consistent with the previous LPS tracing within the first hour after injection (*Figure 2* and *Video 7*), but shown more definitively by high-resolution confocal imaging. By contrast, LPS were not localized within the CNS vasculature using the endothelial reporter *kdrl:mCherry*, but only in the peripheral blood vessels (*Figure 2—figure supplement 2*), indicating that transport of LPS did not likely result from a disruption of the BBB or a direct transit through brain blood vessels. Tracing the movement of LPS from the brain to the periphery also revealed its transient flow through the liver sinusoids prior to immune cell infiltration (*Figure 2*, *Video 5*). However, LPS did not appear to accumulate or bind to cellular structures within the liver as we did not detect LPS there (*Figure 2b*, *Figure 2—figure supplement 2* and *Video 5*), suggesting either the transient exposure of hepatic cells to LPS, or yet unknown extrahepatic signals trigger the hepatic response to systemic LPS. Taken together, LPS appeared to enter circulation via a mechanism directed by the lymphatics for clearing away excess substances from the brain parenchyma and interstitial fluids (*Figure 2e*).

In light of the broad LPS distribution, we sought to functionally test whether the liver response after brain-LPS activation was due to systemic LPS. To create a systemic LPS condition akin to mammalian models of sepsis or endotoxemia (*Szabo et al., 2002*; *Mathison and Ulevitch, 1979*), we directly injected LPS intravenously at the caudal plexus into the bloodstream and compared its resulting peripheral response to that after brain-LPS injection (*Figure 2*). These injections resulted in different outcomes for microglial activation at six hpi, whereby brain-LPS led to a strong activation of microglia but intravenous (IV)-LPS did not (*Figure 2f*). Both routes, however, led to the same robust macrophage infiltration of the liver (*Figure 2g*). To determine if circulation was required for the liver response after brain-LPS injection, we used a morpholino to knockdown *tnnt2*, a cardiac muscle troponin T gene, a well-established reagent for blocking circulation (*Sehnert et al., 2002*; *Figure 2—figure supplement 4*). We indeed found that inhibiting circulation prevented macrophage infiltration into liver after brain-LPS injection (*Figure 2—figure supplement 4*). Taken

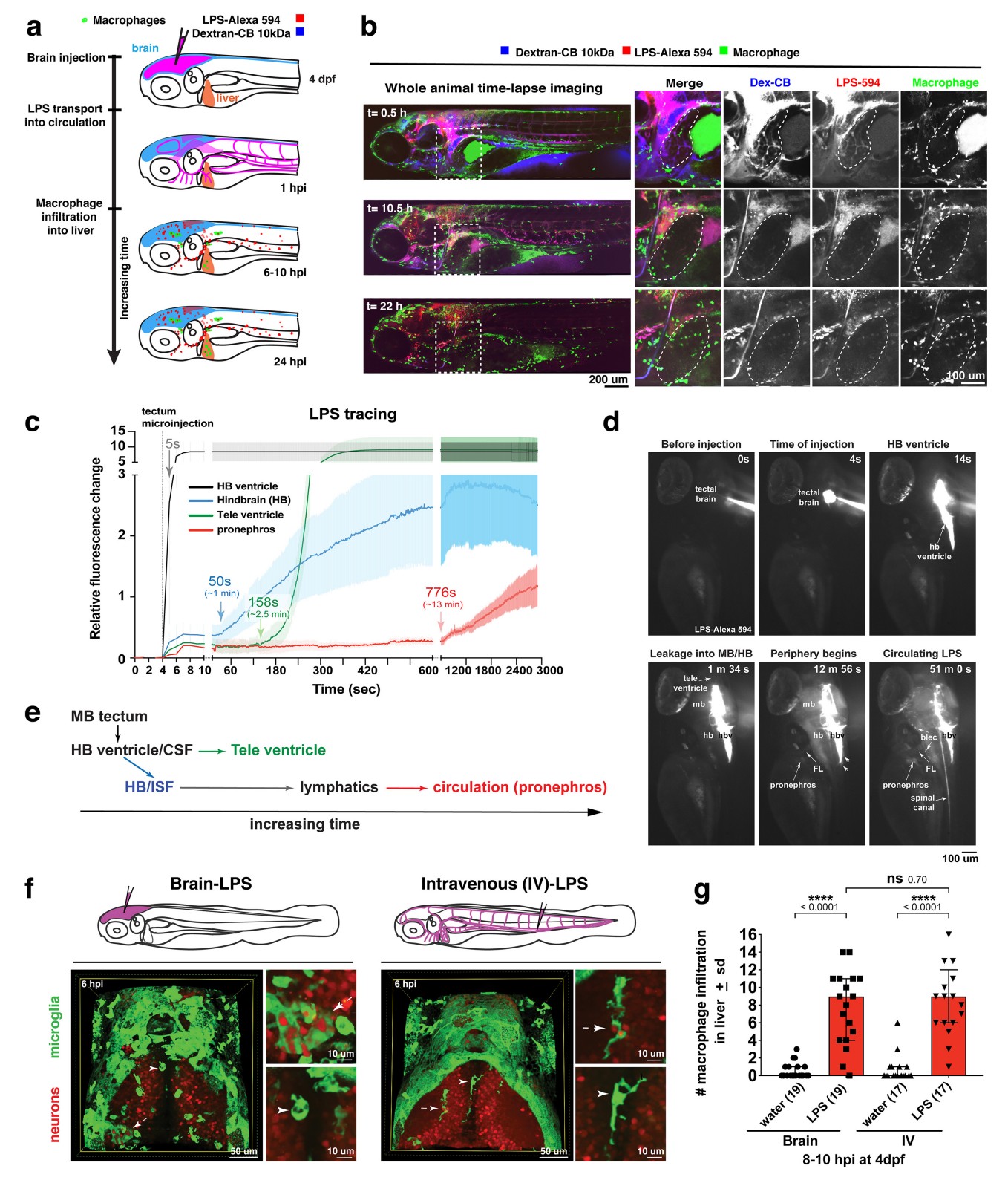

**Figure 2.** Brain-LPS microinjection leads to drainage of LPS molecules into circulation, and causes a hepatic response similar to intravenous LPS injection. (a) Schematic showing time-course of LPS and macrophage distribution. Fluorescently Alexa 594 tagged LPS shown in red, co-tracer dextran-cascade blue shown in blue, and detection of both is shown in magenta. (b) Left, representative 3D images from three timepoints of a 24-hour time-lapse imaging of a large frame stitched from four z-stack tiles (corresponding to *Video 5*). Right, high magnification of the 3D images in dotted box

*Figure 2 continued on next page*

*Figure 2 continued*

region on the left panel showing liver (dotted region) and surrounding area. Merged overlays and individual channels showing Dex-CB (blue), LPS-594 (red), and macrophages (*mpeg1:GFP*+). (c) Live recording of the brain microinjection at four dpf using Alexa 594 or Alexa 488 conjugated LPS was conducted to trace the distribution of LPS in real time at 1 frame per second using an automated acquisition software on a Leica M165 FC stereomicroscope with a high speed and high sensitivity deep-cooled sCMOS camera (DFC9000 GT). Kinetic time plot of relative fluorescence change ± sem of fluorescently tagged LPS starting before the injection at 0 seconds; data from three independent injected animals were used to generate plot. Time of injection was at the 4 seconds timepoint. Arrows indicate the timepoint at which initial LPS signals were detected in the corresponding anatomical location. In some injected animals, LPS also flowed anteriorly from the midbrain ventricle into the telencephalon ventricle starting at about 2.5 minutes after injection (see *Figure 2—figure supplement 3*). (d) Still images representing key events of the dispersion of LPS starting from before to nearly 1 hour after the brain microinjection corresponding to *Video 7*. hb, hindbrain; tele, telencephalon; hbv, hindbrain ventricle; mb, midbrain; FL, facial lymphatics; blec, brain lymphatic endothelial cells; CSF, cerebrospinal fluid; ISF, interstitial fluid. (e) Schematic showing the major route through which LPS were transferred from the site of brain microinjection to peripheral circulation. (f) Top, illustration of brain and intravenous LPS injections. Bottom, 3D tectum brain volume from confocal live imaging at four dpf at 6 hours after brain or intravenous LPS injection using a 40x objective. Microglia (*mpeg1:GFP*+) and surrounding neurons (*nbt:DsRed*+) shown. Small panels show high magnification of microglia (arrows) and neurons corresponding to arrows in the large 3D brain volume image on the left. LPS injection in the brain led to a striking morphological activation of rounded and clustering microglia (arrows), but not by intravenous injection of LPS at 6 hpi. Superficial planes of the head are eliminated to allow visualization of the internal microglia, because the cranial skin surface is highly auto-fluorescent in the GFP channel. (g) Quantification of macrophage infiltration at 8-10 hpi in the four dpf zebrafish larvae. Two-tailed Welch's t-test was used to determine statistical significance. sd, standard deviation; ns, not significant; ****, p<0.0001; LPS-594, LPS-Alexa 594; Dex-CB, cascade blue conjugated dextran, sem, standard error of means. Numbers in parenthesis represent *n*, number of animals analyzed.

The online version of this article includes the following figure supplement(s) for figure 2:

**Figure supplement 1.** Injected LPS macromolecules disperse from the hindbrain ventricle into the hindbrain and spinal cord/trunk interstitial spaces and are localized within the head and trunk lymphatics.

**Figure supplement 2.** Injected LPS macromolecules into tectal brain results in distribution of LPS within the facial lymphatic network and peripheral vasculature.

**Figure supplement 3.** Microinjection of fluorescent LPS into brain tectum at four dpf results in low LPS transfer from hindbrain ventricle into brain interstitial space at around 30% and into the periphery at less than 15% of injected LPS level.

**Figure supplement 4.** Blocking circulation by a morpholino-mediated *tnnt2* knockdown prevented macrophage infiltration into the liver 8 hr after brain injection with LPS.

together, several lines of evidence show that circulating LPS that causes systemic inflammation was driving macrophage infiltration into the liver after brain-LPS injection: 1) systemic distribution of LPS, 2) sufficiency of IV-LPS injection to cause liver infiltration, and 3) blocking circulation prevented macrophage infiltration of the liver.

## Macrophage recruitment to the liver is a MyD88-dependent inflammatory response requiring the IL-34 pathway

We next tested the effects of anti-inflammatory drugs on liver response after brain-LPS microinjection to address whether systemic inflammation was indeed responsible for the macrophage recruitment into the liver. We screened five small-molecule drugs (GW2580, 17-DMAG, celastrol, Bay 11–7082, and dexamethasone) known to effectively curb inflammation by attenuating NF-kB mediated transcription or activating glucocorticoid functions in zebrafish and other systems (*Conway et al., 2005*; *Shimp et al., 2012*; *Sevin et al., 2015*; *Coffin et al., 2013*; *Lancet et al., 2010*; *Venkatesha et al., 2012*; *Yu et al., 2010*; *Aghai et al., 2006*; *Yamamoto and Gaynor, 2001*; *Lee et al., 2012*). These small molecules are known to act through different mechanisms: GW2580, a selective inhibitor of cFMS kinase that blocks the receptor tyrosine kinase CSF1R function which can prevent NF-kB activation (*Caescu et al., 2015*; 17-DMAG (a water-soluble geldanamycin analog) and celastrol, both potent inhibitors of the heat-shock protein Hsp90 that cause disruption or degradation of its target proteins, including the NF-kB protein complex (*Shimp et al., 2012*; *Lee et al., 2006*; Bay 11–7082, an inhibitor of E2 ubiquitin (Ub) conjugating enzymes, which target NF-kB inhibitor, IkB-alpha, for proteasomal degradation (*Lee et al., 2012*; and dexamethasone, an agonist of the glucocorticoid receptor (GR) that activates a negative feedback mechanism to reduce inflammation (*Aghai et al., 2006*). We first assessed the effects these small molecules had on macrophage infiltration into the liver after brain-LPS microinjection using whole-mount RNA in situ hybridization with the macrophage marker *mfap4* (*Figure 3—figure supplement 1*). As positive controls for liver infiltration, we used brain-LPS injected animals that were untreated (water only) for

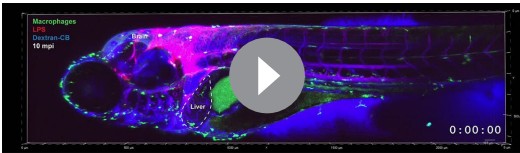

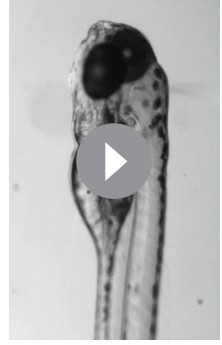

**Video 5.** Time-lapse imaging of whole-body response to LPS microinjection in the brain shows recruitment of macrophages to the liver. 3D images of whole-body from time-lapse imaging of 4 × 200 µm z-stack tiles stitched into one large frame taken at 5 µm z-steps using a Plan Apo lambda 20x objective every 10 min over a 24 hr total period starting at ~15–20 min post brain-LPS injection at four dpf. A sub-volume of the whole stack is shown from 60 µm beneath the most exterior body surface in order to remove obstructive tissue layers blocking the view of the liver. Macrophages are shown by the *mpeg1:GFP* reporter, LPS-Alexa 594 is labeled by red fluorescence, and dextran is visualized by its cascade blue fluorescent tag. Macrophages throughout the body are highly mobile upon brain perturbation, but over time, a number of these cells is found to be restricted inside the liver in contrast to their fast movement through most of other organs and tissues. Fast moving macrophages throughout body were found to mostly express bright *mpeg1:GFP* as opposed to the moderately weaker reporter expression by liver-infiltrating macrophages. See *Figure 2* for additional description of this imaging analysis. Movie file shown at 30 fps.

https://elifesciences.org/articles/58191#video5

**Video 6.** Video showing the initial restriction of brain microinjection to the brain parenchyma, ventricles, and spinal canal using a fluorescent dextran tracer in the four dpf zebrafish. Live recording of the brain microinjection of Alexa 568 conjugated dextran (10 kDa) at faster than video rate (>30 frames per second, fps) using a Leica M165 FC stereomicroscope with a high speed and high sensitivity deep-cooled sCMOS camera (DFC9000 GT). Video shows the left side profile of a live wild-type four dpf zebrafish prior to injection using brightfield imaging followed by the fluorescent dextran injection as shown by the overlay of epi-fluorescence with the brightfield. The fine capillary needle is shown penetrating the injection site in the left brain tectum of the zebrafish. Immediately after injection, the injected substance rapidly fills the brain ventricles that contain the cerebrospinal fluid, and subsequently the central canal of the spinal cord. Video represents a total of 5 s in real time.

https://elifesciences.org/articles/58191#video6

comparison with water-reconstituted drugs (17-DMAG and dexamethasone), and treated with only DMSO for comparison with DMSO-reconstituted drugs (GW2580, celastrol, and Bay 11–7082). Untreated animals without brain microinjection were also used as negative controls. By in situ analysis, we found that Bay 11–7082 and dexamethasone substantially reduced the frequency of macrophage infiltration after brain-LPS injection compared with control groups (*Figure 3—figure supplement 1*). These effects were further validated by in vivo imaging of macrophage recruitment into the liver, which enabled a precise macrophage count at a high cellular resolution (*Figure 3—figure supplement 1*). These results indicated that macrophage infiltration into the liver after brain-LPS stimulation can be prevented by suppressing inflammation via mechanisms inhibiting the NF-kB pathway or activating the glucocorticoid signaling.

To examine possible components affiliated with the NF-kB pathway that may drive macrophage recruitment into the liver, we examined whether an intracellular signal adaptor protein myeloid differentiation protein-88 (MyD88) of Toll-like receptors (TLRs) known for recognizing LPS and mediating cytokine production was essential (*Medzhitov, 2001*). We employed an effective *myd88* specific splice-blocking morpholino to knockdown *myd88* as previously described (*van der Sar et al., 2006*) during the liver response to brain-LPS stimulation (*Figure 3a,b*). The efficacy of *myd88* morpholino to mediate splice-blocking was confirmed by RT-PCR analysis (*Figure 3c*). By in vivo imaging, we found that the number of liver-infiltrating macrophages after brain-LPS microinjection was significantly reduced when *myd88* function was disrupted (*Figure 3a,b*). To further validate these results, we performed the brain-LPS and control injections in *myd88* null mutants and their siblings derived from a heterozygous incross (*Figure 3d*). *myd88* mutants showed either few or no macrophages in the liver after brain-LPS injection at 16–24 hpi similar to baseline brain-water injected animals (*Figure 3d*), demonstrating a much stronger effect in reversing macrophage infiltration than the partial *myd88* knockdown by morpholinos. These results show that the inflammatory liver response

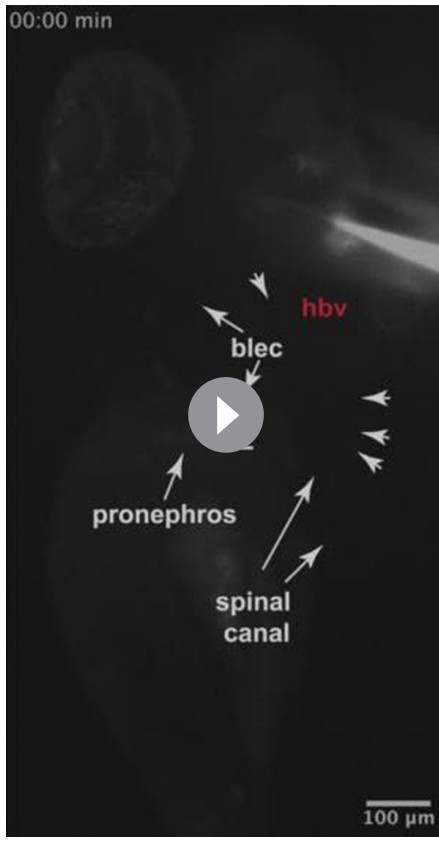

**Video 7.** Rapid in vivo tracking of LPS movement in real time starting before brain tectal injection to nearly 1 hr post injection. Representative live recording of brain microinjection of LPS-Alexa 594 and its immediate aftermath in a four dpf zebrafish larvae at a high temporal resolution at one frame per second (fps) on a Leica M165 FC stereomicroscope with a high speed sCMOS camera (DFC9000 GT). LPS were found to concentrate in the hindbrain ventricle (hbv) immediately after injection and disperse along the ventricular system including the spinal canal. From the hbv, a low level of LPS were exuded out from the posterior end (arrowheads) as well as from the top arms (arrowhead) into the hindbrain interstitial space. Appearance of LPS, albeit weak, can be detected along the facial lymphatics (FL) as well as in the peripheral circulation as represented by the accumulation in the pronephros. Over time, brain lymphatic endothelial cells (blec) were found to accumulate LPS from the brain interstitial fluid. See *Figure 2* and *Figure 2— figure supplement 3* for the kinetic analysis of the LPS tracing, and *Figure 2—figure supplements 1* and *2* for high cellular resolution analysis of LPS localization. Movie file represents a total of 50 min and 41 s of tracking, shown at 300 fps (300x faster than original process).

https://elifesciences.org/articles/58191#video7

depended on the MyD88 pathway and are consistent with LPS-triggered inflammation as the driver of macrophage infiltration into the liver.

As possible hepatic signals that recruit macrophages into the liver during systemic inflammation, we examined whether the interleukin-34 (IL-34) and colony stimulating factor-1 (CSF-1) that share a common receptor CSF1R have a role. IL-34 and CSF-1 are known to mediate various functions of macrophages including inflammatory processes and promoting production of pro-inflammatory chemokines (*Caescu et al., 2015*; *Sauter et al., 2016*; *Masteller and Wong, 2014*). They have recently been shown to be required for macrophage migration and colonization of the brain to form microglia in zebrafish (*Oosterhof et al., 2018*; *Kuil et al., 2019*; *Wu et al., 2018*). Interestingly, engineering an artificial expression of *il-34* in hepatocytes has been shown to be able to recruit macrophages to the liver in zebrafish, but the physiological relevance was not known (*Jiang et al., 2019*). In light of these previous studies, *il-34* and the two zebrafish orthologs of CSF-1 gene (*csf1a* and *csf1b*) were strong candidates for attracting macrophages to the liver after brain-LPS injection. To examine this possibility, we first determined whether these genes (*il-34*, *csf1a*, and *csf1b*) were upregulated in the liver after brain-LPS injection using quantitative PCR (qPCR) analysis on liver-specific and body-minus-liver tissues compared with control animals without the brain microinjection (*Figure 3e*). We found that while *csf1a* and *csf1b* were either not detected or had no difference in the liver with or without LPS injection, *il-34* was significantly upregulated in the liver after brain-LPS microinjection (*Figure 3e*). This upregulation was specific to the liver as *il-34* was not elevated in the body-minus-liver tissue after brain-LPS injection (*Figure 3e*). To test if *il-34* was a required cytokine for recruiting macrophages in our experimental paradigm, we used CRISPR/Cas9 targeted mutagenesis as previously described (*Earley et al., 2018*; *Oosterhof et al., 2018*) to disrupt *il-34* function to assess the immune infiltration (*Figure 3f–g*, *Figure 3—figure supplement 2*). To verify the efficacy of the gene knockdown in transient *il-34* $F_0$ Crispr-injected animals, we determined whether they phenocopied a reduced microglia phenotype recently described in *il-34* stable mutants (*Kuil et al., 2019*). Indeed we found about 40% of the transient *il-34* $F_0$ Crispr-injected animals to have highly decreased microglial numbers (*Figure 3f*). By Sanger sequencing

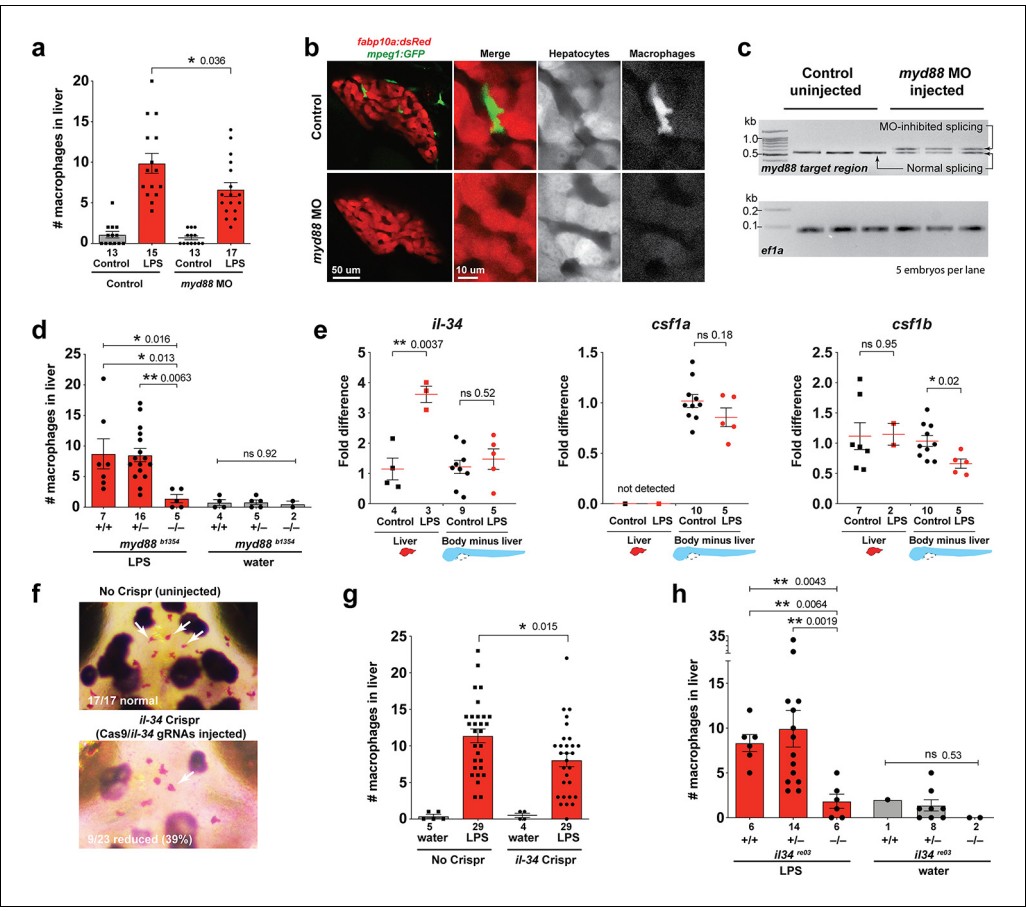

**Figure 3.** Infiltration of liver by macrophages triggered by brain-LPS injection is dependent on adaptor protein *myd88* and cytokine *il-34*. (a) Quantification of macrophage infiltration 6 hr after brain-LPS injection or control treatment (brain-water injection and no injection combined) in *myd88*-deficient morpholino-injected animals compared with control wild-type siblings at four dpf. (b) Left column, representative single plane of whole liver (DsRed+) showing macrophage infiltration (GFP+) in the control animal but not in the *myd88* morpholino-injected animals. Second to fourth columns, high magnification of the merged overlay and single channels showing a single macrophage (GFP+) stationed between hepatocytes (DsRed+) in control but not in *myd88* morpholino-injected animals. (c) RT-PCR analysis showing efficacy of *myd88* morpholino in blocking normal *myd88* splicing at three dpf. Elongation factor one alpha (*ef1a*) PCR used as a sample quality control. (d) Complementary experiments using *myd88* mutants derived from a heterozygous incross show either few or no macrophages in the liver after brain-LPS injection at 16–24 hpi similar to baseline brain-water injected animals, demonstrating a much stronger effect in reversing macrophage infiltration than the partial *myd88* knockdown by morpholinos. (e) qPCR analysis of *il-34*, *csf1a*, and *csf1b* expression in liver only and body-minus-liver tissues comparing brain-LPS injected animals with the control group (brain-water injected and uninjected animals combined) at 6 hpi in four dpf zebrafish. (f) Representative images of control (top) and *il-34* $F_0$ Crispr-injected (bottom). Microglia reduction observed in transient *il-34* $F_0$ Crispr-injected animals shown by neutral red staining (microglia, white arrows), phenocopying previously described stable *il-34* mutants (*Kuil et al., 2019*). (g) Quantification of macrophage infiltration indicates a significant reduction at 8–10 hpi in four dpf transient *il-34* deficient $F_0$ Crispr-injected animals. (h) Stable *il34* mutants derived from a heterozygous incross show either few or no macrophages in the liver after brain-LPS injection at 16–24 hpi similar to baseline brain-water injected animals, showing a much stronger effect in eliminating macrophage infiltration than in the partial gene knockout in transient *il-34* $F_0$ Crispr-injected animals. Statistical significance was determined by a two-tailed t-test coupled with a F-test validating equal variances for two-way comparisons, and Kruskal-Wallis multiple comparisons test for three-way comparisons in d and h (shown by the top bar) followed by corrected two-way tests if the multiple comparisons test was significant. *, p<0.05; **, p<0.01; ns, not significant; data points in scatter plots represent *n*, independent biological samples or animals. Numbers below bar graphs represent *n*.

The online version of this article includes the following figure supplement(s) for figure 3:

*Figure 3 continued on next page*

*Figure 3 continued*

**Figure supplement 1.** Anti-inflammatory drugs dexamethasone and Bay 11–7082 were effective for preventing liver infiltration by macrophages after brain-LPS injection.

**Figure supplement 2.** Characterization of indel mutations in Cas9/*il-34* gRNAs injected animals confirms disruption of the *il-34* coding sequence.

analysis, we verified that these $F_0$ Crispr-injected animals induced a high frequency of frameshift indels altering reading frames and introducing early stop codons in the *il-34* locus (*Figure 3—figure supplement 2*). We found that transient *il-34* $F_0$ Crispr-injected animals after brain-LPS injection indeed had a significantly reduced number of infiltrating macrophages compared with controls (no Crispr injection) after brain-LPS injections (*Figure 3g*). To confirm these results, we also tested the effects of brain injections in stable *il-34* null mutants and their control siblings derived from a heterozygous incross (*Figure 3h*). *il-34* mutants had either few or no macrophages in the liver after brain-LPS injection at 16–24 hpi similar to baseline brain-water injected animals (*Figure 3h*), providing clear evidence that *il-34* is essential in recruiting macrophages to the liver after brain-LPS activation. Taken together, these results showed that macrophage infiltration into the liver was driven by inflammatory processes that require the *myd88* pathway as well as *il-34* signaling likely coming from the liver.

## Liver infiltration by macrophages may be driven and coordinated by neutrophils

Since inflammation is typically a concerted response of the innate immune system, the liver response after brain-LPS stimulation may involve other immune cells besides macrophages. To examine this possibility, we investigated whether neutrophils, the other functional leukocytes prominent at early larval zebrafish stages (*Bennett et al., 2001*; *Xu et al., 2012*), could participate in infiltrating the liver along with macrophages. Using live imaging in transgenic zebrafish at four dpf prior to Kupffer cell development, we quantified the numbers of neutrophils and macrophages in the liver 8–10 hpi with control water or LPS, as well as in uninjected controls (*Figure 4a*). We performed the same brain injection experiments also at 8–10 dpf after Kupffer cell establishment (*Figure 4b*). Our data interestingly showed that neutrophils also significantly infiltrated the liver after brain-LPS but not in the control brain-water and uninjected groups. Similar to macrophages, neutrophils infiltrated the liver irrespective of the presence of Kupffer cells (*Figure 4a–c*). To test whether liver infiltration would still occur at juvenile adult stages, when the liver, blood and lymphatic vasculature, and brain structures are fully mature, we conducted brain tectum injections in 1 month-old zebrafish which had a standard length of 0.8–1 cm (*Figure 4—figure supplement 1*). We found at 16–18 hpi that the numbers of macrophages and neutrophils in the liver were significantly increased after LPS injection compared with water vehicle injections and uninjected controls (*Figure 4—figure supplement 1*), indicating that the liver infiltration persists even at young adult stages. These results implicate that the hepatic response to brain-LPS injection is independent of Kupffer cells, age, or maturity of the brain vasculature and architecture, and drainage of LPS from brain to periphery may still be evident in adulthood. They also raise the possibility that macrophages coordinate with neutrophils for moving into the liver, and signals other than from Kupffer cells can recruit these leukocytes into the liver during systemic inflammation induced by circulating LPS.

To interrogate a possible coordination between macrophages and neutrophils, we turned to gene perturbations to examine the functional consequence of eliminating one immune cell type on the other. We utilized a previously established morpholino targeting *irf8*, an essential transcription factor for macrophage formation, whose effect phenocopies the macrophage-lacking *irf8* mutants for ablating all macrophages at embryonic and early larval stages (*Li et al., 2011*; *Shiau et al., 2015*). Assessment of neutrophil infiltration was conducted at 3–5 hpi, which is earlier than the timepoint of macrophage analysis, because we found neutrophils to infiltrate the liver first before macrophages (*Figure 4l*, *Video 8*). Strikingly, a highly significant 3-fold increase of liver-infiltrating neutrophils was found in macrophage-ablated *irf8* morpholino-injected animals after brain-LPS injection at four dpf compared with the control brain-LPS animals, while brain-water injected controls had no neutrophil infiltration into the liver with or without macrophages (*Figure 4d*). To verify these

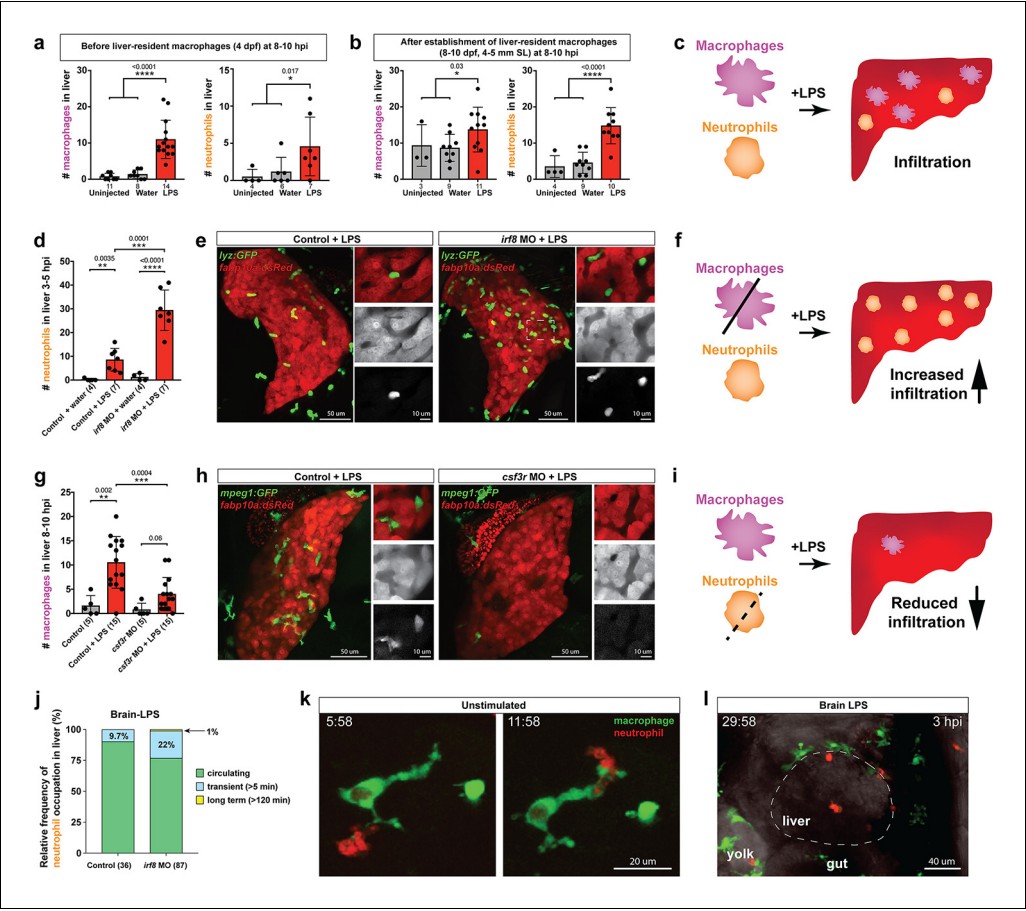

**Figure 4.** Neutrophils and macrophages coordinate to infiltrate the liver during a systemic inflammatory response. (a–b) Total count of macrophages or neutrophils in liver in control (uninjected and brain-water injected) or brain-LPS challenged animals before and after development of liver-resident macrophages (Kupffer cells) at four dpf and 8–10 dpf, respectively. (c) Diagram showing typical immune infiltration after LPS addition in wild-type animals. (d-e) Effects of macrophage ablation by *irf8* knockdown on neutrophil numbers in the liver 3.5–5 hr after brain-LPS injection at four dpf. (d) Quantification of neutrophil numbers. (e) Confocal 3D volume imaging of the whole liver with high magnification of a small region shown on the right that is showing a single z-plane image: top, merged channels; middle, hepatocytes (DsRed+); and bottom, neutrophils (GFP+). (f) Diagram summarizing the effect of macrophage ablation on causing an increase in neutrophil infiltration after LPS injection. (g–h) Depletion of neutrophils using the *csf3r* morpholino reduced macrophage infiltration compared with control LPS injections 8–10 hpi at four dpf. (g) Quantification of macrophage numbers. Significantly fewer macrophages were observed in the liver after neutrophil ablation in brain-LPS injected animals. (h) Same format of images as in e. (i) Diagram summarizing the effect of neutrophil reduction. (j) Comparison of relative frequency of each type of neutrophil occupation in the liver with normal (Control) or depleted (*irf8* MO-injected) levels of macrophages after brain-LPS injection, as determined by in vivo time-lapse imaging. (k) Representative 3D images of normal macrophage and neutrophil interactions around the liver at four dpf (corresponding to *Video 9*). (l) 3D image of macrophage and neutrophil interactions after brain-LPS injection at three hpi in the four dpf larvae showing entry of neutrophils into liver prior to macrophages (corresponding to *Video 8*). Statistical significance was determined by a two-tailed t-test and with Welch's correction for unequal variances as determined by a F-test. MO, morpholino. Each data point in scatter plots represents an independent animal; *n*, number of animals analyzed is shown below each bar graph. Transgenes used: *mpeg1:GFP* for macrophages, *lyz:GFP* for neutrophils, and *fabp10a:DsRed* for hepatocytes.

The online version of this article includes the following figure supplement(s) for figure 4:

**Figure supplement 1.** Hepatic response to brain-LPS challenge still evident after establishment of Kupffer cells and adult vasculature/brain structures in juvenile adults.

**Figure supplement 2.** Clodronate-mediated macrophage depletion results in an increase in neutrophil infiltration after brain-LPS injection.

*Figure 4 continued on next page*

*Figure 4 continued*

**Figure supplement 3.** CSF3R/GCSFR knockdown was effective in reducing neutrophils for revealing neutrophil effects on macrophages.

**Figure supplement 4.** *csf3r* $F_0$ Crispr-injected animals have a significant reduction in macrophage infiltration of liver after brain-LPS injection consistent with results from morpholino-mediated *csf3r* knockdown.

findings, we used a complementary approach for macrophage depletion by using clodronate-containing liposomes to induce apoptosis in macrophages as previously described (*van Rooijen and Hendrikx, 2010*; *Figure 4—figure supplement 2*). Significantly increased neutrophil infiltration of liver after brain-LPS injection in clodronate-mediated macrophage depletion was also found compared with the control response to brain-LPS (*Figure 4—figure supplement 2*). Both methods of macrophage depletion corroborate to show striking increases in neutrophil recruitment to the liver after brain-LPS microinjection. These results suggest that neutrophils may compensate for macrophage loss, or that they are normally constrained by macrophages in part possibly by being phagocytosed as they become impaired or die, as has been described in an immune response (*Prame Kumar et al., 2018*). However, these infiltrating neutrophils do not appear to fully make up for macrophage functions as they do not exhibit the same dynamic movements or lasting occupation in the liver during the hepatic response to brain-LPS (*Figure 4j*). Most of the infiltrating neutrophils are circulating through the liver (>80%) regardless of macrophage presence in the liver during brain-LPS response, albeit a larger fraction is transient (22%) and long-term (1%) after macrophage ablation (*Figure 4j*). Our data therefore favor the latter explanation that macrophages may limit ongoing recruitment of inflammatory neutrophils.

Conversely, we employed a splice-blocking morpholino targeting the granulocyte colony-stimulating factor receptor (CSF3R/GCSFR) for reducing neutrophil numbers in zebrafish (*Pazhakh et al., 2017*). We verified by in situ gene expression pattern of the neutrophil marker, *mpx*, that *csf3r* morpholino-injected animals indeed had neutrophil numbers reduced on average ~30% with a maximum of a 65% decrease (*Figure 4—figure supplement 3*). Also, by in vivo imaging of larval zebrafish carrying both macrophage and neutrophil reporters, the *csf3r* morpholino-injected animals showed no significant change in macrophage number, but a substantial neutrophil reduction (*Figure 4—figure supplement 3*). Although we cannot completely rule out this possibility, these results indicate *csf3r* knockdown likely does not impact baseline macrophages, consistent with neutrophil-specific reduction and functional defects shown in zebrafish *csf3r* mutants (*Pazhakh et al., 2017*; *Basheer et al., 2019*). In these neutrophil-reduced *csf3r* morpholino-injected animals after brain-LPS injection, we found significantly reduced macrophage infiltration of liver by in situ and live imaging analyses (*Figure 4* and *Figure 4—figure supplement 3*). As a complementary approach, we assessed the effect of *csf3r* gene disruption by Crispr injection on macrophage infiltration after brain-LPS injection and also found a significant decrease (*Figure 4—figure supplement 4*). Both morpholino- and $F_0$ Crispr injection- mediated *csf3r* knockdown corroborate to suggest a possible neutrophil role in recruitment of macrophages into the liver during

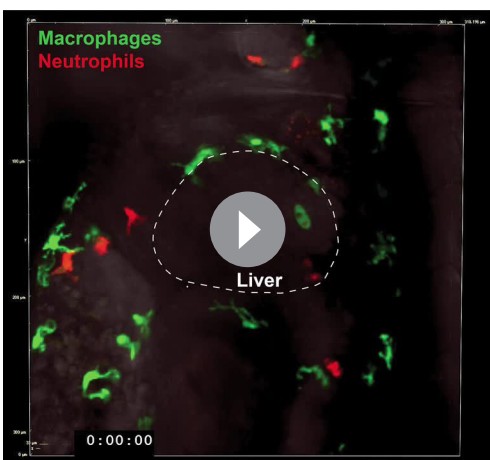

**Video 8.** Dynamic macrophage-neutrophil interactions following brain-LPS microinjection. Confocal time-lapse imaging of a 33 µm z-stack was performed in the region surrounding the liver in double transgenic zebrafish carrying the macrophage (*mpeg1:GFP*) and neutrophil (*lyz:mCherry*) reporters at four dpf after brain-LPS injection. Individual macrophages (GFP+) are prevalent around the liver but has not infiltrated the liver at three hpi. By contrast, a few neutrophils (mCherry+) were seen to enter liver, often passing through. A 40x objective was used to acquire a z-stack every 2 min for a total 1 hr period. Movie file shown at 30 fps. See *Figure 4l* for additional description.
https://elifesciences.org/articles/58191#video8

inflammation. While neither method yields a total *csf3r* elimination to deplete most neutrophils, a stronger hindrance to macrophage infiltration may be possible from a complete *csf3r* depletion. In agreement, constant and dynamic intermingling of these two immune cell types were found around the liver normally (*Figure 4k* and *Video 9*). These results taken together show that during liver inflammation, macrophages and neutrophils coordinate intimately, and macrophages are at least in part driven by neutrophils to enter the liver as they infiltrate the liver first (*Figure 4l*, *Video 8*).

## Immune infiltration of liver promotes inflammation and disrupts liver growth

The functional consequence of macrophage and neutrophil infiltration on the liver during a systemic inflammatory response remains unclear. To examine this, we assessed transcriptional changes comparing immune infiltration and lack thereof in the absence of macrophages and neutrophils. Previous studies have implicated the role of infiltrating innate immune cells in causing liver injury and inflammation (*Karlmark et al., 2008*) but whether this effect is conserved in this model of brain-triggered systemic inflammation is unknown. We examined by qPCR analysis a set of 8 genes after brain-LPS injection with or without innate immune cells compared with the control brain-water injected group to gauge the long-term effect of immune infiltration at 48 hpi. Disruption in liver homeostasis can be defined by an upregulation of genes associated with inflammation (such as pro-inflammatory cytokines *tnfa* and *il1b*). Liver under stress, trauma and inflammation is also characterized by activating an acute phase response (*Ramadori and Christ, 1999*) (such as serum amyloid A (*saa*) and interleukin six signal transducer (*il6st/gp130*)). Disrupted liver can also exhibit altered expression level of genes associated with liver growth and function (such as alpha-2-macroglobulin-like (*a2ml*) (*Hong and Dawid, 2008*), aryl hydrocarbon receptor 1a (*ahr1a*) (*Andreasen et al., 2002*),

and glutathione S-transferase pi 1 (*gstp1*) in zebrafish) (*Abunnaja et al., 2017*). To eliminate myeloid cells, we analyzed mutants with a loss-of-function in an essential myeloid transcription factor *pu.1/spi1b* (*Figure 5*) to eliminate macrophages/microglia and neutrophils (*Li et al., 2011*; *Roh-Johnson et al., 2017*). As a complementary approach, we also used an established translation-blocking morpholino targeting *pu.1/spi1b* (*Figure 5—figure supplement 1*) previously shown to eliminate macrophages/microglia and varying levels of neutrophils depending on dosage (*Li et al., 2011*; *Su et al., 2007*). The efficacy of *pu.1* morpholino was phenotypically validated by a complete loss of brain macrophages (microglia) in a sub-sample of *pu.1* morpholino-injected animals using a neutral red staining assay (n = 4/4) as previously described (*Shiau et al., 2013*). The qPCR results indicated that in the control situation with the full complement of macrophages and neutrophils at two days after brain-LPS injection in the four dpf zebrafish larvae, all genes associated with inflammation and acute phase response gene s*aa* were significantly upregulated, while genes known to affect zebrafish liver development and function were either not changed or modestly increased (*Figure 5* and *Figure 5—figure supplement 1*). By contrast, depletion of myeloid cells in *pu.1* mutants and *pu.1* morpholino-injected animals corroborated to show no significant upregulation in all inflammation genes analyzed (*Figure 5* and *Figure 5—figure*

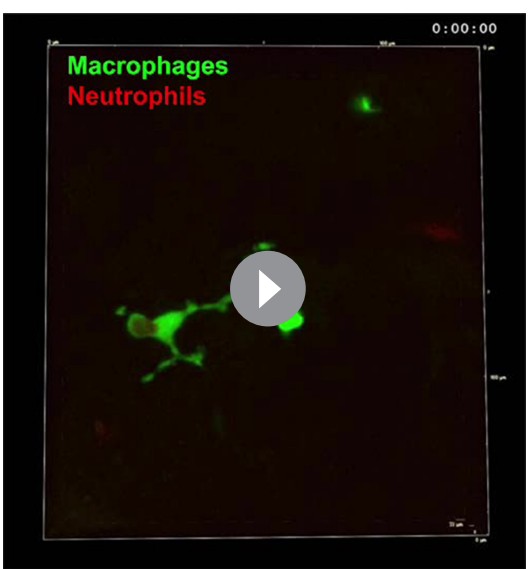

**Video 9.** Dynamic macrophage-neutrophil interactions surrounding the liver normally. Confocal time-lapse imaging of a 33 μm z-stack was performed in the region surrounding the liver in double transgenic zebrafish carrying the macrophage (*mpeg1:GFP*) and neutrophil (*lyz:mCherry*) reporters at four dpf. Individual macrophages (GFP+) and neutrophils (mCherry+) are readily found to intermingle intimately as shown in this video representative. A 40x objective was used to acquire a z-stack every 2 min for a total 1 hr period. Movie file shown at 30 fps. See *Figure 4k* for additional description.

https://elifesciences.org/articles/58191#video9

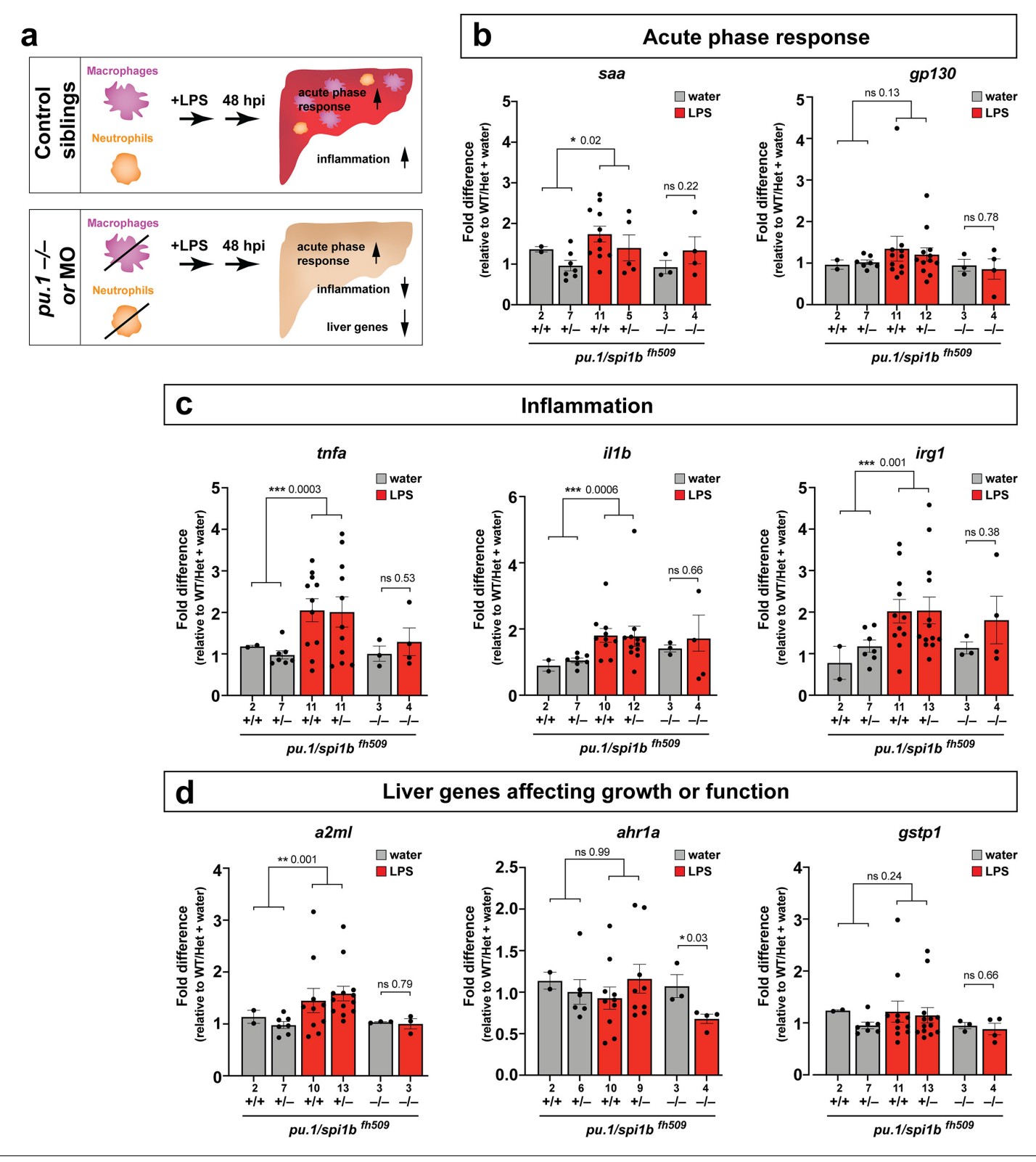

**Figure 5.** Eliminating myeloid cells disrupts hepatic response to systemic inflammation and causes transcriptional programmatic changes. (**a**) Schematic illustrates the impact of myeloid ablation by *pu.1* knockout or knockdown on hepatic response to 48 hr after brain-LPS injection compared with sibling controls. (**b–d**) qPCR was conducted on individual larva 48 hr after either water vehicle or LPS injection in the brain. Brain injections were performed in larvae derived from a *pu.1^fh509* heterozygous incross followed by RNA extraction and genotyping. *pu.1* mutants and their heterozygous and wild-type
*Figure 5 continued on next page*

*Figure 5 continued*

siblings were processed and analyzed in parallel. (**b**) Acute phase response (APR) appears mostly intact in *pu.1* mutants after LPS injection based on a modest elevation of a major APR marker *saa* although not at a significant level. A more significant upregulation was observed in *pu.1* knockdown animals after LPS injection (see *Figure 5—figure supplement 1*). *gp130* is not a specific APR gene and was not significantly changed in all genotypes. (**c**) At 48 hr after brain-LPS injection, relative expressions of all three inflammation genes (*tnfa*, *il1b*, *irg1*) were not significantly upregulated in myeloid-deficient *pu.1* mutants while they remain significantly elevated in control siblings. (**d**) qPCR analysis indicated alteration in two liver-expressing genes affecting zebrafish liver growth or function in myeloid-deficient *pu.1* mutants after brain-LPS injection: *a2ml* was not upregulated, and *ahr1a* was downregulated compared with control siblings, while no significant change was found for *gstp1* in all genotypes. Scatter plots show individual animals; *n*, number of animals analyzed shown below each bar. Statistical significance was determined by a two-tailed t-test and with Welch's correction for datasets with unequal variances. ns, not significant; *, p<0.05; **, p<0.01; ***, p<0.001. See associated data using complementary morpholino (MO) mediated *pu.1* knockdown in *Figure 5—figure supplement 1*.

The online version of this article includes the following figure supplement(s) for figure 5:

**Figure supplement 1.** Transcriptional programmatic changes in *pu.1* knockdown are similar to loss-of-*pu.1* mutants after brain-LPS injection.
**Figure supplement 2.** Serum amyloid A (*saa*), a major acute phase response gene, is not required for liver infiltration by macrophages after brain-LPS activation.
**Figure supplement 3.** Depleting myeloid leukocytes to prevent immune cell infiltration in liver leads to an increase in liver growth during an inflammatory response.

*supplement 1*), indicating that the innate immune cells were responsible for the upregulation of inflammatory genes.

Furthermore, after brain-LPS injection compared with control siblings, *saa* remained mostly upregulated, but *a2ml* and *ahr1a*, two liver genes associated with growth or function had a net reduction in myeloid-deficient animals either by *pu.1* knockout or knockdown (*Figure 5* and *Figure 5—figure supplement 1*). The more significant reduction in liver genes as well as the larger change in acute phase response genes in *pu.1* morpholino-injected animals than in *pu.1* mutants may reflect a speedier recovery from the LPS injection in the mutants as they were not subjected to the early embryonic microinjection as the morpholino animals were. Alternatively, the possibility of not a total but partial depletion of macrophages, neutrophils, or both cell types in *pu.1* morpholino-injected animals may also contribute to differences.

Since *saa* elevation after brain-LPS injection was not eliminated by an absence of macrophages, we further asked whether *saa* may be dispensable for macrophage infiltration. Comparing brain-LPS injections in *saa*^-/-^ mutants side-by-side with injected control siblings showed no difference between the genotypes (*Figure 5—figure supplement 2*), indicating that *saa* is not required for macrophage infiltration into the liver upon LPS activation. The downregulation of liver genes in myeloid cell-depleted animals after brain-LPS injection raised the question as to whether this reflected an actual change in liver growth. To address this, we used in vivo confocal 3D imaging to capture the whole liver at 48 hpi to measure liver size by volume (*Figure 5—figure supplement 3*). We found a decrease in liver size after LPS injection compared with water vehicle injection in both baseline and control-MO groups. Interestingly, by contrast, myeloid-lacking *pu.1* morpholino-injected animals had an increase in liver size after brain-LPS injection (*Figure 5—figure supplement 3*), indicating liver growth was likely impeded by immune cell infiltration during inflammation. Since the expression of some liver genes decreased but the liver size increased in myeloid-deficient animals after brain-LPS injection, these liver genes may not be associated with growth but rather function. These results indicate that the liver infiltration by macrophages and neutrophils promoted inflammation and disrupted liver growth, but had minimal to no effect on the acute phase response gene *saa*, which was not essential for the infiltration (*Figure 5—figure supplement 2*). Taken together, they raise the possibility that immune cell infiltration leads to liver resources and functions being redirected from normal developmental growth to a full-fledged inflammatory response.

## Discussion

### Drainage of macromolecules from brain to body engages the periphery to respond to central changes

As a rapid means by which the brain can engage the periphery in response to possible danger, our study reveals how inflammatory molecules introduced into the brain parenchyma can circulate

outside of the brain to initiate a robust hepatic response in zebrafish (*Figure 6*). It remains to be explored whether the transport of molecules or substances outside of the brain even at trace levels resulting from CNS trauma or injury can explain at least to some degree the associated hepatic damage described in mammals (*Lustenberger et al., 2011*; *Catania et al., 2009*; *Louveau et al., 2016*). The most likely route through which macromolecules can drain out of the brain is through the lymphatic system that removes interstitial fluids, waste products, and immune cells in zebrafish and mammals (*Louveau et al., 2016*; *Jung et al., 2017*; *Okuda et al., 2012*). Recent new knowledge in meningeal lymphatics and glymphatics indicate that macromolecules in the brain can flow from the parenchymal interstitial space into the cerebrospinal fluid (CSF) which drains into the lymphatic vessels or directly into the major veins (sinuses) to enter general circulation (*Louveau et al., 2016*; *Jung et al., 2017*; *Iliff et al., 2012*; *Ma et al., 2017*). In support of this, previous studies using a number of mammalian species (cat, rabbit, dog, and rat) have shown that injection of tracers at 5–100 ul volume such as horseradish peroxidase, India ink or dextran blue into the CSF or brain parenchyma gets drained rapidly into cervical lymph nodes within seconds as well as in entire vasculature within minutes (*Zhang et al., 1992*; *Cserr and Ostrach, 1974*; *Casley-Smith et al., 1976*; *Bradbury and Cole, 1980*; *Bradbury et al., 1981*; *Szentistvanyi et al., 1984*), while radiolabeled albumin microinjected into the brain parenchyma of different brain regions appear rapidly in the CSF and within 4 hr can be found in cervical lymph nodes and the common carotid arteries (*Szentistvanyi et al., 1984*; *Harling-Berg et al., 1989*). Furthermore, metabolites of glucose from the brain are found to be released in the cervical lymph nodes (*Mergenthaler et al., 2013*), supporting the possibility that drainage of endogenous macromolecules from brain to blood circulation can also happen. Taken together, studies in mammals indicate the ability for molecules to transport from the parenchyma or CSF-filled ventricles into general circulation either through the lymphatics, or blood vessels directly. However, the physiological impact of brain drainage on the periphery had not been investigated, nor directly visualized. Our study demonstrates that such drainage can be directly tracked in vivo to induce a robust hepatic response defined by an active recruitment of inflammatory macrophages and neutrophils into the liver.

Similarly, we found that brain tectum microinjection in zebrafish leads to effects comparable to that known of mammalian intracerebroventricular, intracisternal, and intraparenchymal injections (*Glascock et al., 2011*; *Iliff et al., 2012*; *Ma et al., 2017*), such that the injected LPS in the parenchyma rapidly flows into the CSF compartment (or ventricles) likely via bulk flow (*Figure 2* and *Video 7*), followed by clearance of LPS and ISF predominantly by lymphatic drainage (*Figure 2*, *Figure 2—figure supplements 1* and *2*). After brain-LPS injection, we found a low-level spread of LPS/CSF from the hindbrain ventricle into the brain parenchyma and surrounding interstitial space prior to LPS accumulation in brain lymphatic cells and passage through the extracranial lymphatic vessels. Whether CSF flows to the brain more widely than the major perivascular spaces upon intracranial injection remains controversial in mice (*Iliff et al., 2012*; *Ma et al., 2017*), although this appears to occur easily in the larval zebrafish perhaps due to its small brain size conducive to simple diffusion or bulk flow of fluids. Previous work in mice have shown that injected tracers in the subarachnoid CSF or lateral ventricle can enter the brain parenchyma quickly in less than 30 min, and get cleared through paravascular or perineural pathways via the lymphatic system similarly to intraparenchymal injections (*Iliff et al., 2012*; *Ma et al., 2017*), indicating the process of drainage from the brain parenchyma may be conserved between zebrafish and mammals. On average, at about 13 min after tectum microinjection of fluorescent LPS in larval zebrafish, we begin to detect the LPS in circulation by fluorescence, albeit at a baseline intensity, far less than that at the injection site and in the ventricles. This is about two times faster than the transit time for a lateral ventricle injection in mice with a fluorescent tracer to reach blood circulation (around 25 min) (*Ma et al., 2017*), which is still relatively fast. Two key traits that may endow larval zebrafish a faster rate of drainage of solutes from brain ventricle or parenchyma than in rodents may be: 1) the short physical distances between anywhere in the parenchyma and the nearest ventricle and vasculature, which would span no farther than the width of the brain, approximately a few hundred microns, enabling simple diffusion as a primary mode of transport (*Hladky and Barrand, 2018*, and 2) the lack of lymph nodes, based on the prevailing understanding of the zebrafish anatomy (*Yaniv et al., 2006*; *Küchler et al., 2006*), which are lymphatic structures throughout the lymphatic network in mammals that capture and filter the fluid that exits the CNS before joining circulation. The content and concentration of the molecules leaving the CNS that reach blood circulation may be limited by lymph nodes in mammals, but these

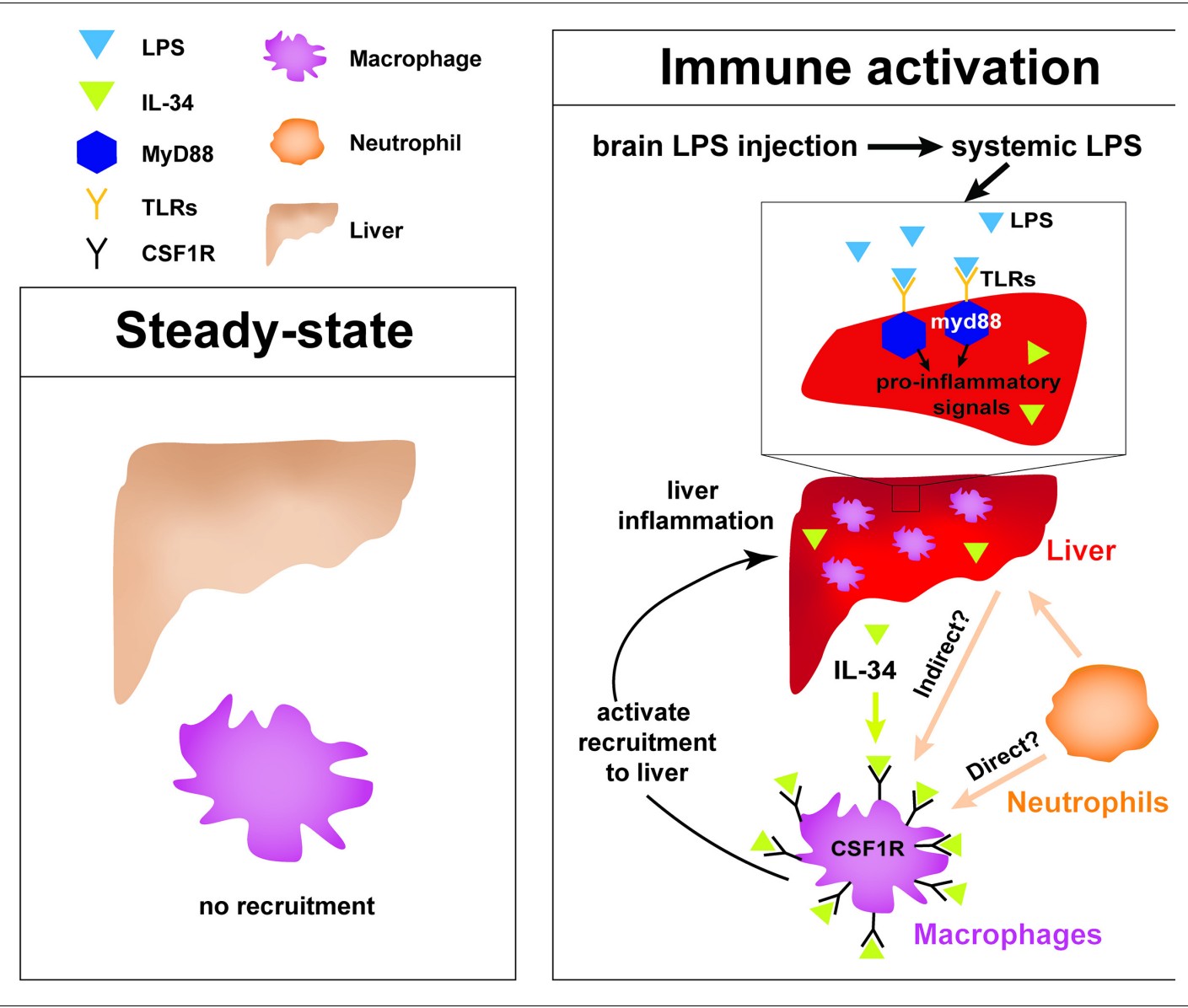

**Figure 6.** Current model for hepatic response to brain drainage of LPS into the periphery. Diagram represents events happening prior to Kupffer cell development in the four dpf zebrafish. Left, at steady-state, peripheral macrophages normally do not migrate into the liver. Right, brain-localized microinjection of LPS leads to systemic distribution of LPS that robustly induces a hepatic response whereby macrophages/monocytes infiltrate the liver. Genes conferring inflammation and acute phase response are highly upregulated. Recruitment of macrophages/monocytes into the liver requires MyD88, a common adaptor protein for TLRs that recognize LPS. Since LPS were found to transiently flow through the liver sinusoids but not appear to accumulate in the liver, it raises the possibility that the requirement for MyD88 may also stem from extrahepatic signals. Although our model illustrates its function only within the liver, whether MyD88 also acts outside of the liver remains to be determined. In addition, IL-34 presumably secreted by the liver and downstream of MyD88 signaling can act as a chemoattractant to macrophages/monocytes expressing CSF1R. Macrophage recruitment may also depend on yet unknown direct or indirect signaling from neutrophils, which infiltrate the liver first subsequent to circulation of LPS.

restrictions may be more lenient in zebrafish due to a possible lack of these structures. Instead of lymph nodes, we found that the intraparenchymally-injected LPS in larval zebrafish after spreading in the brain parenchyma and interstitial space is engulfed by brain lymphatic endothelial cells (blec), which are non-lumenized (*Venero Galanternik et al., 2017*; *van Lessen et al., 2017*; *Shibata-Germanos et al., 2020*), as well as collected by the facial and trunk lymphatic vessels (*Figure 2—figure supplements 1* and *2*). These non-lumenized blec also exist in mouse (*Shibata-Germanos et al., 2020*), but their contribution to the clearance of solutes from the CSF/ISF is yet unclear.

Interestingly, while facial and trunk lymphatic vessels surrounding the brain were filled with LPS by 1.5 hpi, localization of LPS within the brain blood vessels was not observed (*Figure 2—figure supplements 1* and *2*), indicating the main route by which LPS exit the CNS to enter general circulation was via the lymphatic tracts. Additionally, in both zebrafish and mammals, it is also possible that macromolecules to some degree get eliminated by ingestion by perivascular and other phagocytic cells before they drain out to the periphery through the lymphatic-vascular system.

Whether the process of drainage differs during developmental stages when the tissue boundaries and structures of the brain are less mature, we provide evidence that the peripheral hepatic response to brain-LPS injection may be age-independent in zebrafish. The hepatic response was observed from early to late larval stages, and also in juvenile adults, suggesting passage of inflammatory molecules occurs even after establishment of a fully mature brain architecture (including the blood-brain and blood-CSF barriers). This study focused on the early larval stage at four dpf as it offers a timepoint at which infiltrating macrophage cells can be clearly assessed in the liver without the presence of liver-resident macrophages (Kupffer cells). At this age, they are also functionally mature with established functional blood-brain barriers and ventricles with choroid plexus and CSF (*Jeong et al., 2008*; *Henson et al., 2014*; *Fame et al., 2016*). Regardless of the route or quantities at which macromolecules can drain out from the brain, our results implicate the process of drainage from brain as an effective mechanism by which the brain can quickly transmit molecular information to the periphery to engage a prompt hepatic response to potential CNS threats.

## Responsiveness of liver to systemic inflammation irrespective of Kupffer cells

Our data demonstrate that the most striking effect of LPS injection into the brain parenchyma was the recruitment of macrophages into the liver in the zebrafish model (*Figure 6*). In agreement with the liver being most susceptible to systemic inflammation, previous studies using intravenous injection of the endotoxin LPS in different mammalian models have shown an immediate localization of LPS to the liver and less to other organs by tracking radiolabeled LPS (*Bikhazi et al., 2001*; *Mathison and Ulevitch, 1979*; *Braude et al., 1955*; *Herring et al., 1963*). Prior to our work, it was not known that introduction of endotoxin into the brain parenchyma can result in the same impact on the liver as that from a systemic administration. These findings beg the question as to how the liver is the predominant and conserved target of circulating LPS and systemic inflammation in vertebrates from zebrafish to mammals. One overarching explanation could simply be related to the sheer blood volume coming from the hepatic artery and portal vein (*Eipel et al., 2010*) that the liver constantly processes, making it particularly sensitive to inflammatory cues in circulation.

Kupffer cells have been considered to be key immune cells responsible for producing cytokines and chemokines to recruit neutrophils and other leukocytes that may further propagate systemic inflammation (*Bilzer et al., 2006*; *Dixon et al., 2013*). In fact, tracking of radiolabeled LPS has shown that circulating LPS was mostly taken up by Kupffer cells (*Bikhazi et al., 2001*; *Mathison and Ulevitch, 1979*), implicating these cells as the main sink for endotoxins or microbes. By contrast, our study shows that the liver is capable of responding to systemic inflammation and recruiting neutrophils first and then macrophages, independent of Kupffer cells. We found that starting in the larval zebrafish at four dpf when the liver is already functional and differentiated with diverse hepatic cell types including hepatocytes, sinusoidal endothelial cells, biliary cells, and stellate cells (*Yin et al., 2012*), but before Kupffer cells are established, other hepatic cells besides the Kupffer cells are capable of signaling to recruit peripheral leukocytes into the liver. Furthermore, while the gut may provide a source of stimulating molecules through the portal vein, we show that the brain is another source of inflammatory cues that can result in immune infiltration of the liver that was not previously appreciated.

## IL-34 pathway in mediating immune infiltration of liver triggered by systemic inflammation

The receptor tyrosine kinase CSF1R (also known as Fms) pathway regulates migration, differentiation, proliferation, and survival at varying degrees of different tissue-resident macrophages and monocytes in mammals and zebrafish (*Oosterhof et al., 2018*; *Kuil et al., 2019*). Recent studies have implicated a role for this pathway in macrophage recruitment in disease, including the assembly

of tumor-associated macrophages in various cancers (*Noyori et al., 2019*; *Lin et al., 2019*; *Zins et al., 2018*; *Baghdadi et al., 2016*; *Baghdadi et al., 2018*). Of particular interest IL-34 is elevated and implicated in the inflammation process of several inflammatory diseases including rheumatoid arthritis, inflammatory bowel disease, and Sjogren's syndrome (*Ciccia et al., 2013*; *Hwang et al., 2012*; *Moon et al., 2013*; *Chemel et al., 2012*; *Boulakirba et al., 2018*). Interestingly, among the three known ligands of the zebrafish homolog of the receptor tyrosine kinase CSF1R, namely *csf1a*, *csf1b*, and *il-34*, we found that *il-34* was singly upregulated in the liver upon brain-LPS microinjection. Moreover, we showed that disruption of *il-34*, either by antisense splice-blocking morpholinos or a stable loss-of-function mutation, significantly eliminated macrophage infiltration of liver induced by brain-LPS microinjection (*Figure 3*). The requirement for *il-34* signaling presumably from the liver may be coupled with *myd88* known to act downstream of TLR signaling upon LPS recognition, as *myd88* was found to be highly expressed in the normal larval zebrafish liver (*Koch et al., 2018*). Whether these pathways act only in the liver, other cell types, or both, and whether other signaling mechanisms are involved, including CXCR3-CXCL11 known to affect macrophage chemotaxis to infection sites in human and zebrafish (*Torraca et al., 2015*), remain to be explored. Our data indicate an essential role for *il-34* in recruitment of macrophages into the liver, implicating a new function for the *il-34/csf1r* pathway in conjunction with *myd88* dependent mechanisms during hepatic inflammation.

## Coordination of the innate immune system during hepatic response to systemic inflammation

Chronic active infiltration of leukocytes into the liver can cause liver damage and subsequent progression to fibrosis, cirrhosis or liver cancer (*Karlmark et al., 2008*; *Mossanen and Tacke, 2013*; *Huang et al., 2016*). This is a common feature shared among liver diseases caused by infection, toxic insults, or autoimmunity (*Edwards and Wanless, 2013*). Using zebrafish, we found that macrophages and neutrophils were co-dependent in the process of infiltrating the liver after brain microinjection of LPS. We further showed by in vivo time-lapse imaging an early recruitment of neutrophils in the first few hours was followed by migration or circulation of peripheral macrophages into the liver. This is consistent with previous studies showing early neutrophil recruitment followed by monocyte-derived macrophages in other contexts of inflammation (*Soehnlein et al., 2009*; *Jenne et al., 2018*; *Kim and Luster, 2015*). While secreted granule proteins from neutrophils that have already infiltrated the liver may directly recruit macrophages (*Gregory et al., 2002*; *Soehnlein et al., 2008*) in our experimental platform of liver infiltration, we cannot exclude the possibility that indirect effects by which neutrophils alter vascular permeability or signaling from endothelial or other hepatic cell types actually direct macrophage recruitment into the liver.

Conversely, the impact of monocytes and macrophages on neutrophil activity remains less understood. Using two complementary approaches to deplete macrophages either by *irf8* genetic deficiency or liposomal clodronate treatment in zebrafish, we found that macrophages may be important to limit neutrophil infiltration into the liver upon brain drainage of LPS. An excessive level of neutrophil infiltration may lead to a heightened release of toxic metabolic products and proteolytic enzymes (*Prame Kumar et al., 2018*), thereby causing more damage to the liver. Macrophages in the liver are known to remove apoptotic or impaired neutrophils by phagocytosis, which in turn can modulate their own signaling based on the receptor(s) used during neutrophil clearance (*Gregory et al., 2002*). Since deficiency in *irf8* causes an elimination of macrophages but also an increase in baseline neutrophil number (*Shiau et al., 2015*), we cannot completely rule out that a larger neutrophil pool may contribute to the significant increase in neutrophil infiltration after brain-LPS injection in *irf8* morpholino-injected animals. Nonetheless, the agreement in effect from two distinct methods of macrophage ablation, and the observed intimate macrophage-neutrophil contacts in the hepatic region strongly support reciprocal interactions between macrophages and neutrophils.

## Perspective on the brain-liver connection

Our study shows that the drainage following microinjection of LPS into the brain robustly caused peripheral macrophages and monocytes to infiltrate the liver, a process mediated by an upregulation of the cytokine IL-34 in the liver presumably downstream of activating the Toll-like receptor adaptor

protein MyD88, and signaling from neutrophils. Treatment with a subset of anti-inflammatory drugs indicates that glucocorticoid activity or inhibiting NF-kB activation suppresses immune cell infiltration during hepatic response to systemic inflammation. These results prompt intriguing questions on how a hepatic response to a central disruption may reflect the presence and severity of CNS inflammation; whether a hepatic response contributes to the recovery or disruption of brain homeostasis; and to what extent does drainage of harmful or inflammatory molecules from brain to circulation explains for the peripheral problems associated with primarily CNS pathologies. LPS were used as an effective tool for brain-localized microinjection in this study to allow tracing of inflammatory molecules originating in the brain to assess peripheral consequences, but whether endogenous proteins and other molecules could be secreted by the brain and drained into circulation to cause similar effects as the LPS do on the liver remains to be investigated. Some examples that lend support to this possibility relate to hallmarks of Alzheimer's disease that has a known association with liver dysfunction, such as brain tau secretion and its presence in the CSF as well as clearance of toxic accumulations of brain amyloid-beta (*Pernègre et al., 2019*; *Bacyinski et al., 2017*; *Nho et al., 2019*); however the link between the brain secretion or drainage and liver disruption is yet unclear. Understanding mechanisms regulating immune infiltration of the liver and brain-periphery interactions can offer new approaches for modulating or detecting central inflammation while limiting damage elsewhere in the body.

## Materials and methods

### Zebrafish

Embryos from wild-type (TL and AB), mutant and transgenic backgrounds: *il34^re03* (*Kuil et al., 2019*), *myd88^b1354* (*Burns et al., 2017*), *saa^rdu60* (*Murdoch et al., 2019*), *pu.1/spi1b^fh509* (*Roh-Johnson et al., 2017*), *mpeg1:EGFP* (*Ellett et al., 2011*), *lyz:GFP* (*Hall et al., 2007*), *lyz:mCherry; cmlc2:GFP* (*Meireles et al., 2014*), *fabp10a:DsRed* (*Dong et al., 2007*), *kdrl:mCherry-CAAX* (*Fujita et al., 2011*), *mrc1a:egfp^y251* (*Jung et al., 2017*), *mfap4:tdTomato* (*Walton et al., 2015*), and *nbt:DsRed* (*Peri and Nüsslein-Volhard, 2008*) were raised at 28.5°C and staged as described (*Kimmel et al., 1995*). Stable transgenic *mpeg1:BFP* and *fabp10a:BFP* fish lines were generated using Tol2-mediated transgenesis based on cloned constructs combining the BFP coding sequencing (gift from Martin Distel) downstream of their respective regulatory sequences as published for *mpeg1(1.86 kb)* (*Ellett et al., 2011*) and *fabp10a(2.8* kb) (*Murdoch et al., 2019*). This study was carried out in accordance with the approval of UNC-Chapel Hill Institutional Animal Care and Use Committee (protocols 16–160 and 19–132).

### LPS and bacteria microinjections

Zebrafish larvae at 4–10 dpf and 1 month old were mounted dorsal side up for brain injections or on their sides for intravenous injections in 1–3% low-melting agarose. A pneumatic microinjector (WPI) with a fine capillary glass pipette was used to inject 1 nL for 4–10 dpf or 2 nL for 1 month old of stimulus into the targeted site (either tectum for brain injections, or caudal vein plexus for intravenous injections). Lipopolysaccharides (LPS) derived from 0111:B4 *E. coli* (L3024 Sigma) at 5 ng/nL, ultra-pure water, or live *Escherichia coli* cells were supplemented with fluorescently labeled dextran (Invitrogen, 10,000 MW at 1:100 dilution of a 5 ng/nL stock) for visualization. *E. coli* cells were prepared for injections as previously described (*Earley et al., 2018*). Fluorescently tagged Alexa 594 conjugated LPS from 055:B5 *E. coli* (L23353 Sigma) at 5 ng/nL was used for brain microinjection at 1 nL per fish. All fish after microinjection are carefully monitored for normal health and behavior, and nearly all injected fish remain healthy and viable, showing no signs of overt change. These healthy post-injection fish are used for further experimentation and analysis.

### In vivo time-lapse and static confocal imaging

All time-lapse and static z-stack imaging were performed using a Nikon A1R+ hybrid galvano and resonant scanning confocal system equipped with an ultra-high speed A1-SHR scan head and controller. Images were obtained using an apochromat lambda 40x water immersion objective (NA 1.15) or a plan apochromat lambda 20x objective (NA 0.75). Z-steps at 1–2 μm were taken at 40x and 3–5 μm at 20x. Different stages of zebrafish were mounted on glass-bottom dishes using 1.5%

low-melting agarose and submerged in fish water supplemented with 0.003% PTU to inhibit pigmentation. Dissected juvenile and adult liver tissues were mounted in fluoromount-G (Southern Biotech) for imaging.

## Liver cell counts

At larval stages 4–10 dpf, the whole liver in the live transgenic larvae was captured in a z-stack that was used for counting total number of macrophages and neutrophils in the liver. Transgenic reporters labeling both the immune cells and hepatocytes were used to count cells through the optical sections. At the juvenile and adult stages, cell counts were made on whole liver dissected and imaged ex vivo using a maximum intensity projection of the z-stack using the ImageJ cell counter tool. Two representative z-stacks were taken for each liver counted.

## Clodronate-mediated macrophage depletion

Transgenic larvae were injected at three dpf intravenously with 1 nL clodronate liposomes (Liposoma) supplemented with Alexa 568 conjugated dextran (10 kDa, Invitrogen) used at 1:100 for visualization of injection. Larvae were incubated for two days to allow macrophage depletion to occur before they were subjected to brain microinjection with LPS or water vehicle at five dpf. Clodronate depletion of macrophages in the brain (microglia) was confirmed in a subset of larvae using the neutral red staining assay. After brain injections, analysis of neutrophil numbers in the liver was conducted in live transgenic zebrafish imaging at 3.5 hpi.

## Whole mount RNA in situ hybridization

RNA in situ was performed using standard methods. Antisense riboprobes were synthesized from plasmids as described (*Shiau et al., 2013*) encoding *mfap4*, *mpx* (Open Biosystems clone 6960294), and a 739 bp coding fragment of *irg1* (NM_001126456.1; pCES161) cloned from a cDNA library derived from a four dpf *E. coli* injected larva using primers Forward-5'- TCGTTCTGCCAGTAGAGA TGTTA-3' and Reverse-5'- GCGAGCTGAGATGCCTCTAAAC-3'.

## RNA isolation, qPCR and RT-PCR

RNA was isolated following the RNAqueous-Micro kit RNA Isolation Procedure (Ambion). Whole larvae or dissected livers and remaining body were lysed in 100–300 uL RNA lysis buffer. Larval liver dissections were performed on transgenic larvae *Tg*(*fabp10a:DsRed*) to aid in identifying the liver. cDNA was made from 150 or 200 ng of total RNA using oligo (dT) primer with SuperScript IV reverse transcriptase (Invitrogen) for qPCR or RT-PCR analysis. qPCR was performed on the QuantStudio 3 Real-Time PCR System (Applied Biosystems) using SYBR Green. The delta-delta ct method was used to determine the relative levels of mRNA expression between experimental samples and control. *ef1a* was used as the reference gene for determination of relative expression of all target genes. Primer sequences for qPCR and RT-PCR analysis are listed in *Supplementary file 1*.

## Morpholino injections

Antisense morpholino oligos were purchased from Gene Tools and re-suspended in water to make 1 mM or 3 mM stocks. Morpholino sequences are listed in *Supplementary file 1*. Morpholinos were heated at 65°C for 5 min and cooled to room temperature before injecting into single-cell embryos at 0.5–1 nL.

## Neutral red staining

Microglia were scored in live larvae by neutral red vital dye staining as previously described (*Shiau et al., 2015*; *Shiau et al., 2013*). In brief, 3–4 dpf larvae were stained with neutral red by immersion in fish water supplemented with 2.5 µg/mL neutral red and 0.003% PTU at 28.5°C for 1 hr, followed by 1–2 water changes, and then analyzed 2–3 hr later using a stereomicroscope.

## CRISPR-Cas9 targeted mutagenesis of *il-34* and *csf3r*

The target genes were *il-34* (NCBI accession: NM_001128701.1; Gene ID: 560193) and *csf3r* (NCBI accession: NM_001113377.1; Gene ID: 100134935). Co-injection of Cas9 mRNA and guide RNAs (gRNAs) was conducted in wild-type 1-cell stage zebrafish embryos. Cas9 mRNA was transcribed

from XbaI linearalized pT3TS-nCas9n plasmid (Addgene #46757) using mMessage mMachine T3 Kit (Ambion) according to the manufacturer's instructions. CRISPR targets for gRNA designs were identified using CHOPCHOP (http://chopchop.cbu.uib.no) (*Gagnon et al., 2014*). Gene-specific oligonucleotides using T7 promoter were used to make gRNAs as previously described (*Gagnon et al., 2014*). gRNA target sequences and genotyping primers are provided in *Supplementary file 1*. In vitro transcription of gRNAs from assembled oligonucleotides was conducted using the HiScribe T7 Quick High Yield RNA Synthesis Kit (NEB). To ensure high mutagenesis rate and large deletion mutations, three gRNAs were simultaneously injected with Cas9 mRNA for each gene. Injected clutches of embryos were validated to contain CRISPR mediated mutagenesis by a T7 endonuclease assay. Mutations were analyzed by TOPO TA cloning followed by Sanger sequencing.

## Small-molecule anti-inflammatory drugs

Administration of different small-molecule chemicals, DMSO control, or no treatment were performed in parallel starting at three dpf through the time of brain microinjection at four dpf in clean multi-well dishes. Analysis of the effect on macrophages was performed at eight hpi either by RNA in situ hybridization or by live confocal imaging using the liver and macrophage transgenes (*fabp10a:DsRed* and *mpeg1:GFP*). 17-DMAG (5 µM) and dexamethasone (6.5 µM) were reconstituted in water and these treatments were compared to the water controls. GW2580 (25 µM), Celastrol (0.22 µM) and Bay 11–7082 (1 µM) were resuspended in DMSO so these treatments were compared to the DMSO-treated controls. List of small molecules is detailed in *Supplementary file 1*.

## Liver size measurement

Whole liver was imaged 48 hr after LPS or water injection at four dpf in the brain tectum on a confocal Nikon A1R+ using an apochromat lambda 40x water immersion objective (NA 1.15). Four dpf larvae for the brain injections were derived from either uninjected, *pu.1*-MO injected, or negative control *p53*-MO injected wild-type embryos. Z-steps were taken at 1 µm thickness. Surface and 3D rendering of the z-stack to measure the liver volume was conducted using the Imaris 3D/4D Image Analysis Software.

## Statistical analysis

Unpaired two-tailed t-tests were performed unless otherwise noted. F test was used to compare variances. For unequal variances, Welch's correction was used on the two-tailed t-test. For multiple comparisons of 3 or more groups, one-way ANOVA test was applied followed by pair-wise tests to determine the pair(s) showing significant differences. GraphPad Prism eight was used to run statistical tests and create graphs. Scatter bar plots show symbols representing biological replicates.

## Acknowledgements

We thank Jonathan Gable for critical insights and discussions on our manuscript. We are grateful for the generous sharing of fish lines from Tjakko van Ham, John Rawls, David Tobin, and Brant Weinstein, and constructs from Martin Distel. We especially thank the rapid shipment of *spi1b* adult mutant line from Cecilia Moens and Rachel Garcia. We also thank the UNC Neuroscience Microscopy Core Facility for assistance with the Imaris imaging analysis and UNC Zebrafish Aquaculture Core Facility for zebrafish housing and care. L.Y. was funded by a UNC SURF fellowship, and JAJ was supported by a T32 ES007126 training fellowship. This work was supported by NIH NIGMS grant 1R35GM124719 to CES.

## Additional information

### Funding

| Funder | Grant reference number | Author |
| --- | --- | --- |
| National Institutes of Health | 1R35GM124719 | Celia E Shiau |
| National Institutes of Health | T32 ES007126 | Jessica A Jiménez |

| University of North Carolina at Chapel Hill | UNC SURF fellowship | Linlin Yang |

The funders had no role in study design, data collection and interpretation, or the decision to submit the work for publication.

## Author contributions
Linlin Yang, Formal analysis, Investigation imaging/morpholino/RNA in situ/drugs test/time course/qPCR, Visualization, Methodology, Writing - review and editing; Jessica A Jiménez, Formal analysis, Investigation morpholino/imaging/adult Kupffer/time course/qPCR, Methodology, Writing - review and editing; Alison M Earley, Formal analysis, Investigation CRISPR/liver size/morpholino/cDNA/transgenic lines generation, Visualization, Methodology, Writing - original draft, Writing - review and editing; Victoria Hamlin, Victoria Kwon, Investigation genetic mutants and qPCR, Writing - review and editing; Cameron T Dixon, Investigation in situ/imaging, Writing - review and editing; Celia E Shiau, Conceptualization, Formal analysis, Supervision, Funding acquisition, Investigation drainage/mutants/imaging, Visualization, Methodology, Writing - original draft, Writing - review and editing

## Author ORCIDs
Linlin Yang (iD) https://orcid.org/0000-0001-5602-4157
Alison M Earley (iD) http://orcid.org/0000-0003-1889-9221
Celia E Shiau (iD) https://orcid.org/0000-0002-9347-9158

## Ethics
Animal experimentation: This study was performed in strict accordance with the recommendations in the Guide for the Care and Use of Laboratory Animals of the National Institutes of Health. All of the animals were handled according to approved institutional animal care and use committee (IACUC) protocols (#16-160 and #19-132) of the UNC Chapel Hill.

## Decision letter and Author response
Decision letter https://doi.org/10.7554/eLife.58191.sa1
Author response https://doi.org/10.7554/eLife.58191.sa2

# Additional files

## Supplementary files
• Supplementary file 1. Key Reagents used.
• Transparent reporting form

## Data availability
All data generated or analyzed during this study are included in the manuscript and supporting files.

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
