## [Decision Letter]

**Acceptance summary:**

In the manuscript by Yang et al. the authors use a LPS injection paradigm to investigate how the leakage of pro-inflammatory signals from the brain affect peripheral tissues. Using live imaging and genetic manipulations they reveal that following injection of labeled LPS in the brain robust infiltration of macrophages occurs in the liver. They show this response involves IL-34 and MyD88 and is coordinated with neutrophils. This work illustrates system wide responses to inflammatory stimuli and interactions between organs via the circulation of pro-inflammatory signals.

**Decision letter after peer review:**

[Editors’ note: the authors submitted for reconsideration following the decision after peer review. What follows is the decision letter after the first round of review.]

Thank you for submitting your work entitled "Drainage of inflammatory macromolecules from brain to periphery targets the liver for macrophage infiltration" for consideration by *eLife*. Your article has been reviewed by three peer reviewers, including Jean-Pierre Levraud as the Reviewing Editor and Reviewer #3, and the evaluation has been overseen by a Senior Editor. The following individuals involved in review of your submission have agreed to reveal their identity: Helen Stolp; Jean-Pierre Levraud.

Our decision has been reached after consultation between the reviewers. Based on these discussions and the individual reviews below, we regret to inform you that your work will not be considered further for publication in *eLife*.

The question of interplay between CNS and peripheral inflammation is important, and a systemic imaging approach, as zebrafish allows, appears well suited. Live imaging experiments are of high quality and the measurements of macrophage infiltration in the liver were compelling. However, it is unclear if needle-mediated LPS injection in the brain of a larva with a not fully mature BBB is properly modelling neurological injury. The mechanism of LPS drainage from the brain, at the very least, requires better characterization. The analysis of the molecular and cellular pathways involved in liver infiltration was also not fully convincing, with modest changes and improper controls.

Reviewer #1:

The study by Yang, Jimenez, Earley, Dixon and Shiau, titled "Drainage of inflammatory macromolecules from brain to periphery targets the liver for macrophage infiltration" utilises innovative molecular biological techniques in zebrafish to show that inflammatory mediators drain from the brain to the periphery affecting macrophage and neutrophil infiltration in the liver. There are many aspects of this work that are of interest, but the whole work seems a bit confused.

This work seems to mix two important points. The first is how brain inflammation signals to the liver and the second is age specific liver recruitment of macrophages and neutrophils in response to inflammation, and the signalling mechanisms involved in this process. The Introduction, such as it is, addresses this first point, suggesting that liver damage is a common occurrence in conjunction with inflammatory brain injury or degeneration.

However, the authors show the movement of labelled LPS and tracer molecules through the brain interstitial fluid, to the CSF and then on to the periphery within a short period following brain injection. They also show that a systemic LPS injection causes the same response in the liver, and that blocking circulation before doing brain injections inhibits the response. Together these data imply that LPS is being removed from the brain, reaching the periphery and then stimulating its response. While this is interesting, it is not clear that this same pathway exists in normal neurological injury or equivalent interventions in the mammalian brain, where injected substances are found to stay within the parenchyma, and therefore it is unclear how translatable their findings are to other species, or how meaningful they are for an normal injury paradigm. It seems that these authors are studying macrophage and neutrophil recruitment to the liver in the context of peripheral inflammation – it is not clear that the brain has any specific involvement.

Additionally, the authors are using a very large dose of endotoxin, that possibly models meningitis (at best), so if inflammation is usually passed from the brain to the periphery via this pathway (CSF and lymphatics), it is likely to be to a much less extend that what is being investigated here. This work also does not mention the blood brain barrier in the zebrafish, its stage of development and any potential response to inflammation of the brain barriers or CSF production/flow that might confound their results. The authors did not address any potential confounder of the young animals used in this study.

The second half of this work, looking at the differential liver response to systemic inflammation (whether it comes via the brain or not) is more interesting, but still has some significant flaws in its current form.

The myd88 and IL-34 experiments were interesting in showing some involvement of these signalling molecules/systems, however these did not have a large effect, suggesting other signalling systems are also likely to be involved, and the modification of expression was generalised, so the importance of the brain or liver (or other structure) was not investigated.

While the use of animals with manipulation of their macrophages is interesting, the authors appear to have validated the models by assessing microglial numbers in the brain, rather than cell number in the periphery. Given the different dynamics of macrophage turn over in the brain compared to the periphery, and the lack of any clear central contribution to the results presented in this study, it is not clear that this is a meaningful validation of the models (e.g. PU-1, IL-34).

The authors also write about the liver response to inflammation as if it is a surprise, rather than one of the major known functions of the liver. That being said, the use of the developing system does show some really interesting results regarding neutrophil and macrophage recruitment to the liver, and these could be the focus of a very interesting paper if followed up appropriately. The idea that the macrophage/neutrophil recruitment is differentially related to functions such as cytokine production, the acute phase response and factors related to liver health and function is certainly interesting. Could authors produced data on liver growth from their imaging studies to support the idea raised in Figure 5?

Reviewer #2:

In this manuscript the authors aim to investigate the link between brain pathology and liver inflammation/damage. For this they use the zebrafish as a model system, which is well suited to analyze such responses systemically. Much of the work is quite interesting and well supported by high quality in vivo imaging data and whole mount in situ's analyses.

As very apparent from the title and Discussion section, the main point the authors want to make is not that systemic LPS/*E. coli* can cause influx of macrophages, and neutrophils, in the liver, which was quite thoroughly investigated. Instead the authors appear most interested in the drainage of bacterial material from the brain. Some of the main conclusions were not really investigated in my opinion. I think there could potentially be several other reasons for the occurrence of LPS material in the blood. Even though brain vasculature of zebrafish at this early embryonic stage have some properties of a mature blood brain barrier, i.e. expression of occludins/claudins and exclusion of dyes with certain molecular size, it is very immature and could permit passage of LPS not by a specific route by diffusion. Additionally, the brain is punctured by a relatively large needle, which could somehow induce leakage. Or perhaps the pressure of the injection somehow forces the LPS into the blood stream. I guess the 10kd dextran data shows that at least molecules of this size don't exit the brain, suggesting the BBB is not massively disrupted. The injection and the LPS likely also cause cell death in the brain and indeed it seems microglia contain cell debris. Could this somehow contribute to cause the drainage? If the brain drainage is to be the main conclusion of this work, I think this would require more experimental support. E.g. perhaps look also at later developmental stages whether LPS still drains.

Another question is whether the LPS once in circulation accumulates in the liver, causing the attraction of macrophages, or if the influx is caused otherwise. The LPS could also be taken up in endothelial cells or some other vascular cell type, followed by attraction of leukocytes. Please comment on this.

The cause of the increased neutrophil infiltration could be related not to the absence of macrophages but as a side-effect of the macrophage depletion (irf8 loss of function skews differentiation towards neutrophils and clodronate causes death of macrophages, very likely followed by attraction of neutrophils by the corpses if they're not cleared properly).

The authors conclude that glucocorticoid activity suppresses the hepatic response to systemic inflammation. The fish were kept in a quite high dose of dexamethason (100 uM) for more than a day. The fish are rapidly developing at this stage, and dex could affect several mechanisms depending on glucocorticoid activity, including the response of the leukocytes. Therefore, I don't understand what the authors base this conclusion on.

Reviewer #3:

This is an interesting work, mostly based on elegant live imaging. However, if the fact that injection of LPS injection in the brain induces infiltration of macrophages in liver is convincingly established, much of the rest of the analysis is not so strong.

Essential revisions:

1) Morpholino experiments: there are no control morphants, in any experiment! This is particularly critical for Figure 3A where the difference is really small and could be a MO injection side effect.

2) Same problem with il34 CRISPR experiments: the comparison should be made with control crispants injected with an irrelevant sgRNA+CAS9 (alternatively, and even better, a stable il34 mutant could be used)

3) Is there a general increase in macrophage numbers (ie, emergency myelopoiesis) consecutive to LPS injection? This has to be measured and compared with the increase in the liver.

4). Figure 1—figure supplement 1: the strong, compact mfap4 signal seen in the liver after LPS injection is not consistent with the relatively modest macrophage infiltration (no more than 20 cells) seen by live imaging. Could hepatocytes be induced to express mfap4 in these conditions? There are mfap4 reporter lines (to label macrophages), so that would be easy to test.

5) Regarding the terminology "Kupffer cells", applied to liver macrophages, a more cautionary tone is needed throughout the text and figures. Localisation in the sinusoids is generally considered a feature defining Kupffer cells. It would also be worth citing some of the ancient but extant literature that established that most teleost fish have very few Kuppfer cells (e.g., Sakano and Fujita, 1982)

6) Figure 2: an assessment of the fraction of intracranially-injected LPS that leaks to the periphery is necessary.

7) The brain injection causes a temporary hole in the BBB. Can one exclude that this is how LPS reaches the periphery?

8) Figure 4D: since total neutrophil numbers increase in irf8 morphants, it is not clear if this result is significant. This measurement should be somehow normalized to the total neutrophil count.

9) Figure 4—figure supplement 3: a 27% reduction in neutrophil numbers is rather low. Csf3r ablation is also known to have some effect on macrophage populations (Liongue et al., Blood 11::2535, 209). A quantification on the impact of macrophages is required here.

10) The tnnt2a MO experiment is not conclusive, since circulation was blocked from the onset, resulting, at 4dpf, in a smaller brain, a strong edema, and probably general inflammation. Drug- mediated inhibition of heartbeat, or physical blockade of bloodflow, would be more convincing.

[Editors’ note: further revisions were suggested prior to acceptance, as described below.]

Thank you for submitting your article "Drainage of inflammatory macromolecules from brain to periphery targets the liver for macrophage infiltration" for consideration by *eLife*. Your article has been reviewed by Didier Stainier as the Senior Editor, a Reviewing Editor, and three reviewers. The reviewers have opted to remain anonymous.

The reviewers have discussed the reviews with one another and the Reviewing Editor has drafted this decision to help you prepare a revised submission.

In the manuscript by Yang et al., the authors investigate in zebrafish the effect of leakage of pro-inflammatory LPS form the brain in peripheral tissues. Using beautiful live imaging and genetic manipulations, they find that macrophages infiltrate the liver and their recruitment depends on myd88 and il34, thus underscoring the existence of rapid communication between the brain and the liver that may play a role in immune surveillance.

Summary:

Two new reviewers and one reviewer that evaluated the previous version of this work agreed in that the manuscript has been greatly improved with substantial new data and an in-depth revision. Two key experimental manipulations, the knock down of myd88 and il34, are now backed by stable loss of function mutations and several experiments have been strengthen with new or improved analyses. However, there are still two important experimental manipulations using morpholinos that have not been properly controlled. In addition, editorial changes are needed to better explain the use of LPS injections as an experimental tool.

Essential revisions:

1) The authors use morpholinos targeting csf3r and pu.1 expression and draw important conclusions based on those experiments. Given the inherent problems of morpholinos, particularly for inflammation studies, it is necessary to support the use of those reagents with stable mutants and/or additional controls. If mutants are not available or cannot be generated, the knockdown experiments may be further supported with rescue experiments and or F0 Crispr, in which case the significance of any findings related to those experiments should be tempered with an appropriate discussion of the caveats.

2) While brain injections of LPS can be a useful tool as used in this work, it is hardly a physiological condition. An editorial revision should address caveats and limitations, perhaps highlighting the use of this experimental approach as a tool.

---

## [Author Response]

[Editors’ note: the authors resubmitted a revised version of the paper for consideration. What follows is the authors’ response to the first round of review.]

The question of interplay between CNS and peripheral inflammation is important, and a systemic imaging approach, as zebrafish allows, appears well suited. Live imaging experiments are of high quality and the measurements of macrophage infiltration in the liver were compelling. However, it is unclear if needle-mediated LPS injection in the brain of a larva with a not fully mature BBB is properly modelling neurological injury. The mechanism of LPS drainage from the brain, at the very least, requires better characterization. The analysis of the molecular and cellular pathways involved in liver infiltration was also not fully convincing, with modest changes and improper controls.Reviewer #1:The study by Yang, Jimenez, Earley, Dixon and Shiau, titled "Drainage of inflammatory macromolecules from brain to periphery targets the liver for macrophage infiltration" utilises innovative molecular biological techniques in zebrafish to show that inflammatory mediators drain from the brain to the periphery affecting macrophage and neutrophil infiltration in the liver. There are many aspects of this work that are of interest, but the whole work seems a bit confused.This work seems to mix two important points. The first is how brain inflammation signals to the liver and the second is age specific liver recruitment of macrophages and neutrophils in response to inflammation, and the signalling mechanisms involved in this process. The Introduction, such as it is, addresses this first point, suggesting that liver damage is a common occurrence in conjunction with inflammatory brain injury or degeneration.

Thank you for the suggestion—we have now expanded the introduction to include a better discussion of the context from which we are addressing how the brain may relay information to the liver. We are interested to explore a possible route through which the brain may affect liver function simply by a drainage of pro-inflammatory substances or molecules, albeit even at a trace level, to instigate systemic inflammation, rather than a direct brain to liver signaling. Regarding the second point, our results do not indicate age-specific recruitment of leukocytes into the liver. We have now included new additional data showing juvenile adults (Figure 4—figure supplement 1) as well as late-stage larvae at 8-10 dpf (Figure 4B), when BBB/brain structures are mature, show liver infiltration consistent with the earlier four dpf larval stage that our study focuses on.

However, the authors show the movement of labelled LPS and tracer molecules through the brain interstitial fluid, to the CSF and then on to the periphery within a short period following brain injection. They also show that a systemic LPS injection causes the same response in the liver, and that blocking circulation before doing brain injections inhibits the response. Together these data imply that LPS is being removed from the brain, reaching the periphery and then stimulating its response. While this is interesting, it is not clear that this same pathway exists in normal neurological injury or equivalent interventions in the mammalian brain, where injected substances are found to stay within the parenchyma, and therefore it is unclear how translatable their findings are to other species, or how meaningful they are for a normal injury paradigm.

There has been a substantial number of studies in mice and other mammalian species (including recent works reviewed by (Hladky and Barrand, 2018), Fluids Barriers CNS and conducted by Ma et al., 2017; Ma et al., 2019; Iliff et al., 2012) that lay out the different evidence for drainage or passage of tracers and molecules from the CSF or parenchyma to blood circulation via the lymphatic system or the venous blood directly through the arachnoid projections. Previous work in mice and other mammalian species have shown that injected tracers in the subarachnoid CSF or lateral ventricle can enter the brain parenchyma quickly in less than 30 minutes, and get cleared through paravascular or perineural pathways via the lymphatic system similarly to intraparenchymal injections (Iliff et al., 2012; Ma et al., 2017). These studies provide the groundwork to suggest that our findings related to brain drainage in zebrafish may be broadly relevant to other species.

We have improved the discussion of these points with new data and revised text. New experiments were conducted to provide a more detailed analysis of tracing LPS directly (Figure 2A-B, Figure 2—figure supplement 1, Figure 2—figure supplement 2, Figure 2—figure supplement 3 and Video 7) as well as text that include more discussion of previous studies in other species in the Discussion section: “Drainage of macromolecules…” Overall, our study aims to address fundamental questions relating to brain-periphery interaction rather than any specific brain injury or disease model, although results of this work could have relevance to these conditions.

It seems that these authors are studying macrophage and neutrophil recruitment to the liver in the context of peripheral inflammation – it is not clear that the brain has any specific involvement.

At the outset of our study it was not known whether direct signaling between the brain and the liver would occur upon injection of inflammatory activators into the brain; our data support the conclusion that drainage of LPS from the brain into circulation occurs and is sufficient to explain liver inflammation and immune cell recruitment. We therefore focus our analysis and interpretation on the consequences of such transport of molecules from brain to periphery, which addresses an important area yet under-explored and under-appreciated. While much research has focused on mechanisms of penetrating the blood-brain barrier (BBB) or the blood-cerebrospinal fluid (BCSF) barrier to enable entry of a peripheral agent into the brain parenchyma such as those relating to infectious diseases causing brain dysfunction (for example, by neurotropic viruses including *Rabies lyssavirus*, West Nile virus, and cytomegalovirus) and drug delivery to the brain (Hladky and Barrand, 2016; van den Pol, 2009), far less attention has been given to investigating the reciprocal transfer from brain to circulation and its consequences. Therefore, we think our study has significant implications for future work.

Additionally, the authors are using a very large dose of endotoxin, that possibly models meningitis (at best), so if inflammation is usually passed from the brain to the periphery via this pathway (CSF and lymphatics), it is likely to be to a much less extend that what is being investigated here. This work also does not mention the blood brain barrier in the zebrafish, its stage of development and any potential response to inflammation of the brain barriers or CSF production/flow that might confound their results. The authors did not address any potential confounder of the young animals used in this study.

The threshold at which an endotoxin or another inflammatory substance could cause a liver response in other species is an interesting point, however this goes beyond the scope of our current study.

We agree that a discussion on the brain barriers (BBB and BCSF) would be helpful for our interpretation. We should add that it is known that by four dpf (the stage of our experimentation), the BBB has been shown to be established based on Claudin-5 and ZO-1 expression in the cerebral microvessels as well as restricted permeability to HRP and other tracers (Jeong et al., 2008), as well as the choroid plexus/ventricle flow (Henson et al., 2014; Fame et al., 2016). We have now included new data examining the blood and lymphatic brain vasculatures (Figures 2—figure supplement 1 and Figure 2—figure supplement 2) that suggest the passage of LPS relies largely on the lymphatic and ISF in the brain rather than the brain blood vessels, before entering peripheral circulation. We have also added text in the Introduction, Results section and Discussion section that discuss these points.

The second half of this work, looking at the differential liver response to systemic inflammation (whether it comes via the brain or not) is more interesting, but still has some significant flaws in its current form. The myd88 and IL-34 experiments were interesting in showing some involvement of these signalling molecules/systems, however these did not have a large effect, suggesting other signalling systems are also likely to be involved, and the modification of expression was generalised, so the importance of the brain or liver (or other structure) was not investigated

Since morpholino-based gene knockdown does not completely eliminate the target gene function and can yield partial effects, we have now added new data using stable null mutants which do not have a functional target gene. The results significantly strengthen our previous findings that myd88 and il34 mechanisms are essential for liver infiltration. The effects were clear-cut. Both il34 and myd88 null mutants exhibited no liver infiltration after brain-LPS injection, rather than merely reduced as in the morphants. We added these data in Figure 3D and 3H. Based on our data showing liver specific gene upregulation of il34 after brain-LPS injection (Figure 3) and the known endogenous expression of myd88 in the normal larval liver (Koch et al., 2018), they suggest that myd88/IL-34 mechanism functions in the liver. However, we do agree that the specific cell type(s) and region(s) that require these mechanisms still remain to be fully determined, and it is possible that multiple cell types rely on these pathways to coordinate the liver infiltration. We have added this point to the Discussion section.

While the use of animals with manipulation of their macrophages is interesting, the authors appear to have validated the models by assessing microglial numbers in the brain, rather than cell number in the periphery. Given the different dynamics of macrophage turn over in the brain compared to the periphery, and the lack of any clear central contribution to the results presented in this study, it is not clear that this is a meaningful validation of the models (e.g. PU-1, IL-34).

In zebrafish, LOF mutations in these genes (Pu.1, irf8, and il-34) directly affect microglia development during early embryonic and larval stages. Furthermore, the peripheral macrophage number is directly linked with brain microglial number because these peripheral macrophages in part seed the brain to form microglia. For example, il-34 is required for migration and colonization of peripheral macrophages in the brain, and thus a loss of microglia is a direct readout for disruption of il34 function (Kuil et al., 2019; Wu et al., 2018), whereas phenotype in the peripheral macrophages is not clear-cut due to the intrinsic variation in macrophage cell number between individuals. Loss of peripheral macrophages leading to a complete absence of microglia has also been directly shown in spi1b/pu.1 morphants/mutants (Li et al., 2011; Xu et al., 2015) and irf8 morphants/mutants (Shiau et al., 2015). Therefore, a readout of a loss of microglia has been used to effectively and efficiently demonstrate and validate a loss of function in these genes.

The authors also write about the liver response to inflammation as if it is a surprise, rather than one of the major known functions of the liver. That being said, the use of the developing system does show some really interesting results regarding neutrophil and macrophage recruitment to the liver, and these could be the focus of a very interesting paper if followed up appropriately. The idea that the macrophage/neutrophil recruitment is differentially related to functions such as cytokine production, the acute phase response and factors related to liver health and function is certainly interesting. Could authors produced data on liver growth from their imaging studies to support the idea raised in Figure 5?

As for clarification, we do not think a response by the liver to systemic inflammation is surprising, but the underlying mechanisms that make this organ a usual suspect are interesting and poorly understood. In particular, how might leukocyte infiltration in the liver contribute to the hepatic response. To address an aspect of this, we think the direct visualization of macrophage/neutrophil infiltration during a hepatic response, which had not been conducted prior to our work, offers the much-needed insights to begin to understand this process.

We agree the differential effects were interesting. As suggested, we have followed up on this result to address whether an absence of immune infiltration affects liver growth after brain-LPS challenge in pu.1 morphants. We used two separate cohorts of controls animals (baseline and control-MO injected WT groups) to determine typical liver growth after 48 hpi comparing LPS injection to water vehicle injection. Interestingly, we found that instead of a reduction in liver size as one might predict from a reduction in liver genes expression, liver sizes increased in LPS injected pu.1 morphants, indicating liver growth was likely impeded by immune cell infiltration during inflammation. We have added new data as presented in Figure 5—figure supplement 1 and relevant text in the Results section. The summary schematic in Figure 5A has also been updated with the new result reflected.

Reviewer #2:In this manuscript the authors aim to investigate the link between brain pathology and liver inflammation/damage. For this they use the zebrafish as a model system, which is well suited to analyze such responses systemically. Much of the work is quite interesting and well supported by high quality in vivo imaging data and whole mount in situ's analyses.

Thank you for recognizing the value and interest in this work.

As very apparent from the title/discussion, the main point the authors want to make is not that systemic LPS/*E. coli* can cause influx of macrophages, and neutrophils, in the liver, which was quite thoroughly investigated. Instead the authors appear most interested in the drainage of bacterial material from the brain. Some of the main conclusions were not really investigated in my opinion. I think there could potentially be several other reasons for the occurrence of LPS material in the blood. Even though brain vasculature of zebrafish at this early embryonic stage have some properties of a mature blood brain barrier, i.e. expression of occludins/claudins and exclusion of dyes with certain molecular size, it is very immature and could permit passage of LPS not by a specific route by diffusion. Additionally, the brain is punctured by a relatively large needle, which could somehow induce leakage. Or perhaps the pressure of the injection somehow forces the LPS into the blood stream. I guess the 10kd dextran data shows that at least molecules of this size don't exit the brain, suggesting the BBB is not massively disrupted. The injection and the LPS likely also cause cell death in the brain and indeed it seems microglia contain cell debris. Could this somehow contribute to cause the drainage? If the brain drainage is to be the main conclusion of this work, I think this would require more experimental support. E.g. perhaps look also at later developmental stages whether LPS still drains.

These are all good points. We have conducted several new experiments to provide a more thorough characterization of the brain LPS microinjection, in order to provide full confidence of our brain-specific injection method, and to clarify the process of brain drainage of LPS. To do this, we analyzed the detailed kinetics of LPS movement starting from the localized brain tectum injection using a high-speed and high-sensitivity stereoscope cMOS capture at 1 Hz (see new data added in Figure 2A-B, Figure 2—figure supplement 1, Figure 2—figure supplement 2, Figure 2—figure supplement 3, and Video 7) and the same brain injected animals were subjected to confocal imaging at later timepoints to characterize localization of LPS relative to blood and lymphatic vessels (see new data added in Figure 2—figure supplement 1, Figure 2—figure supplement 2). While most of the LPS remained in the ventricular system after tectal injection, we found the hindbrain ventricle to be the key region from which LPS appeared to leak out into interstitial spaces. Interestingly, LPS accumulated within the brain lymphatic endothelial cells (blec) as well as the facial and trunk lymphatic vessels as shown by co-localization of LPS within the lymphatic reporter *mrc1a:GFP* but *not* within the brain vasculature as marked by *kdrl:mCherry* (see Figure 2—figure supplement 1, Figure 2—figure supplement 2), indicating that the presence of peripheral LPS did not likely stem from a disruption of the BBB or a direct passage through brain blood vessels. We also do not see a typical microglial response at the time of LPS passage to indicate a link between these or neuronal cells contributing to drainage, see Figure 2—figure supplement 1, Figure 2—figure supplement 2 where microglia are labeled. We generated a new *mpeg1:BFP* line in order to visualize LPS/blood or lymphatic vasculature/microglia in 3-colored fish. Instead, LPS appeared to enter circulation via a mechanism directed by the lymphatics for clearing away excess substances from interstitial fluids.

These results are now provided as multiple new data files: Figure 2A-B, Figure 2—figure supplement 1, Figure 2—figure supplement 2, Figure 2—figure supplement 3 and Video 7. We are replacing the previous Figure 1—figure supplement 5 on the drainage and passage of dextran with this new large data set, because the tracing of dextran does not directly nor accurately reflect LPS movement, and the use of a manual image acquisition at only one frame every minute is too slow to yield reliable and clear tracing of the dye. Due to the nature of the previous imaging setup, we also resorted to using a more blunt-ended needle for better visualization purposes than the fine-tipped needle we use for our routine brain microinjections (as shown in Figure 2A-B and Video 7). Our new tracing analyses address LPS drainage directly, and the movie acquisition has significantly improved using an automated capture at a very rapid speed of 1 Hz (60x faster than previous study) for about 1 hour of continuous imaging.

In terms of the concern of the maturity of the brain structures at the stage of our experiments at 4dpf, we have extended our studies into older stages when the brain structures (including BBB) are fully mature. We conducted experiments in 1 month old juvenile adults, which still showed significant increases in both macrophage and neutrophil numbers in the liver after brain-LPS injection (see Figure 4—figure supplement 1), as well as at late larval stages at 8-10 dpf after several days of feeding, substantial presence of Kupffer cells and BBB maturation that the hepatic response persists after brain-LPS injection (see Figure 4B). These results indicate that the hepatic response to the brain perturbation is independent of age or maturity of the brain vasculature and architecture, and drainage of LPS from brain to periphery may still be evident even in young adults.

Finally, we should add that our data corroborate previous data shown by others indicating that by four dpf the BBB and brain ventricles are well established with boundaries and restrictions (Jeong et al., 2008; Henson et al., 2014; Fame et al., 2016); what we have observed for LPS passage at four dpf using our injection method matches a stereotypical pathway in every fish we analyzed and does not appear to be a random or broad leakage or diffusion process that allows LPS to get out to the periphery.

Another question is whether the LPS once in circulation accumulates in the liver, causing the attraction of macrophages, or if the influx is caused otherwise. The LPS could also be taken up in endothelial cells or some other vascular cell type, followed by attraction of leukocytes. Please comment on this.

Good question. Tracing the movement of LPS from the brain to the periphery revealed its transient flow through the liver sinusoids prior to infiltration by macrophages/neutrophils (Figure 2, Video 5). However, LPS did not appear to accumulate or bind to cellular structures within the liver as we did not detect LPS there (Figure 2B, Figure 2—figure supplement 2 and Video 5), suggesting either the transient exposure of hepatic cells to LPS, or yet unknown extrahepatic signals trigger the hepatic response to systemic LPS. Although the liver endothelial cells were not found to bind to LPS, other major peripheral blood vessels do, such as the posterior cardinal vein (PCV), which we show in Figure 2—figure supplement 2. This could certainly be a source of cells that could attract macrophages, but how macrophages are signaled to enter the liver where there is no direct LPS accumulation remains an open question. Text has been added in the results to point to these different possibilities.

The cause of the increased neutrophil infiltration could be related not to the absence of macrophages but as a side-effect of the macrophage depletion (irf8 loss of function skews differentiation towards neutrophils and clodronate causes death of macrophages, very likely followed by attraction of neutrophils by the corpses if they're not cleared properly).

These are great points. We had the same concerns so (1) the analysis of control brain-water injections in both irf8 and clodronate based depletions, and (2) use of 2 complementary approaches (irf8 LOF and clodronate) based on very different mechanisms and distinct caveats were critical to provide interpretable results if they both show the same effect. If there were dramatic “side-effects” we expected to see significant baseline neutrophil infiltration in the absence of LPS, but there was not (Figure 4D and Figure 4—figure supplement 2). Examining clodronate-induced macrophage effects in our own hands, we only found partial depletion of the macrophage population, and the remaining macrophages (arrows) either appear morphologically normal or take on a ball-shape but not appearing inflammatory or clustering in groups of corpses, suggesting an unlikely scenario for a mass immune response (representative data provided on right for your reference).

Although irf8 knockdown causes a significant increase in baseline numbers of neutrophils, we did not observe an apparent increase in neutrophils surrounding the liver after LPS compared with controls (Figure 4E and data not shown), and furthermore, the shift in neutrophil behaviors after LPS towards more “macrophage-like”, taking on a more transient and long-term association with the liver stood out from those infiltrating in controls (with normal macrophages) (Figure 4J). These traits suggest distinct changes in neutrophils that may not be explained merely by an increased baseline number. Nonetheless, we cannot entirely rule out the possibility that an increased baseline neutrophil number could contribute to the increased liver infiltration, so we have added text to address this point in the Discussion section. We believe the agreement in the result from these two different methods logically lend support to this interpretation of the data, although the potential caveat should be stated.

The authors conclude that glucocorticoid activity suppresses the hepatic response to systemic inflammation. The fish were kept in a quite high dose of dexamethason (100 uM) for more than a day. The fish are rapidly developing at this stage, and dex could affect several mechanisms depending on glucocorticoid activity, including the response of the leukocytes. Therefore, I don't understand what the authors base this conclusion on.

Upon reviewing this point, we revisited our documentation, and realized that the water-soluble form of dex we used does not have the normal dex MW of 392.46 g/mol, because the vendor provides this reagent at 65 mg of dex per gram due to cyclodextrin used to balance the chemical (which is in a side note that we unfortunately had missed); product information can be found at https://www.sigmaaldrich.com/catalog/product/Σ/D2915?lang=en&region=US. We very much thank the reviewer for pointing out this issue, which led us to catch this mistake in a timely manner. All places reporting this concentration has been corrected to show 6.5 uM.

We agree that it is possible that the glucocorticoid activity could be directly affecting the immune cells, liver cells, or both. We have modified the text to specify that it suppresses the macrophage infiltration, which our data shows (Figure 3—figure supplement 1), rather than specifying which cell type it may affect. In the Discussion section, the text now reads: “Treatment with a subset of anti-inflammatory drugs indicates that glucocorticoid activity or inhibiting NF-κB activation suppresses immune cell infiltration during hepatic response to systemic inflammation.”

Reviewer #3:This is an interesting work, mostly based on elegant live imaging. However, if the fact that injection of LPS injection in the brain induces infiltration of macrophages in liver is convincingly established, much of the rest of the analysis is not so strong.

Thank you for recognizing this work to be of interest.

Essential revisions:1) Morpholino experiments: there are no control morphants, in any experiment! This is particularly critical for Figure 3a where the difference is really small and could be a MO injection side effect.

Point well taken; we now have included mutants to address these concerns for analysis of myd88 and il34 effects, see new data in Figure 3. The mutants resulted in strong and clear-cut reversal of macrophage infiltration providing evidence that both mechanisms are required.

2) Same problem with il34 CRISPR experiments: the comparison should be made with control crispants injected with an irrelevant sgRNA+CAS9 (alternatively, and even better, a stable il34 mutant could be used)

Point well taken. We have now addressed this using stable il34 mutants which became available recently. See Figure 3H.

3) Is there a general increase in macrophage numbers (ie, emergency myelopoiesis) consecutive to LPS injection? This has to be measured and compared with the increase in the liver.

Yes, there appears to be a general increase in overall macrophage number after LPS injection, as expected. However, there is no actual increase in the liver to speak of, because at baseline (uninjected) without LPS there are effectively zero macrophages normally in the liver at four dpf (Figure 1A, Figure 4A); so we are looking at an induction of macrophages in the liver rather than an “increase”. This is the important reason for using four dpf for most of our experiments as it provides a time window when baseline is effectively zero, and any presence of a macrophage in the liver usually marks an extraordinary event due to an immune response. Based on the behaviors of the infiltrated macrophages after LPS injection, a substantial fraction of them (~13%) entered and remained in the liver for a long period of time (several hours)(Figure 1—figure supplement 5), suggesting their actions were not random and fleeting merely due to a general cell number increase, but rather tailored for the liver.

4). Figure 1—figure supplement 1: the strong, compact mfap4 signal seen in the liver after LPS injection is not consistent with the relatively modest macrophage infiltration (no more than 20 cells) seen by live imaging. Could hepatocytes be induced to express mfap4 in these conditions? There are mfap4 reporter lines (to label macrophages), so that would be easy to test.

Good point. First, a point of clarification, we have seen at 4dpf brain-LPS injections that the range of infiltrated macrophage numbers can go well above 20, and also above 30 per liver which is substantial considering the small number of total cells in the liver in larval zebrafish (Figure 1C, Figure 3D, 3G, 3G, Figure 4A), and that normally it would take an age of 14 dpf to reach numbers of ~ 20 resident macrophages per liver (Figure 1A). However, we were also curious about the reason why in some cases of the mfap4 in situ after brain-LPS injection there was seemingly a broad liver expression. We have added the following text and data provided in Figure 1—figure supplement 3 to address these points.

“Due to some examples of broad liver *mfap4* in situ expression (Figure 1—figure supplement 1) in comparison to a discrete number of infiltrating macrophages by live imaging after brain-LPS injection, we assessed whether this could be explained by an induction of ectopic *mfap4* expression in the liver upon LPS activation. Using a transgenic line *mfap4:tdTomato* to mark cells expressing the *mfap4* gene, we found *mfap4* restricted to macrophages and absent in liver cells (Figure 1—figure supplement 3), suggesting the broad liver *mfap4* expression may be diffuse in situ signals coming from a liver more densely populated by infiltrated macrophages.”

5) Regarding the terminology "Kupffer cells", applied to liver macrophages, a more cautionary tone is needed throughout the text and figures. Localisation in the sinusoids is generally considered a feature defining Kupffer cells. It would also be worth citing some of the ancient but extant literature that established that most teleost fish have very few Kuppfer cells (e.g., Sakano and Fujita, 1982)

Ultimately, Kupffer cells are defined by their macrophage origin and normal residence in the liver, namely “liver-resident macrophage”. We appreciate the suggestion to make our definition more clear so we have modified the text at the start of the relevant Results section.“To relate macrophage presence in the liver after brain-LPS injection to macrophages that normally reside in the liver (which we refer to as Kupffer cells), we analyzed the developmental timing of Kupffer cells, which had not been previously described, starting at the 4 dpf larval stage to the 40 dpf juvenile adult stage (Figure 1A).” Because in vivo imaging of Kupffer cells has remained scant/absent, we think the dynamic nature of these cells to inform their localization and relationship with the sinusoids remains a very open question and poorly explored in all species. Sakano and Fujita, 1982 used ultrastructural microscopy to examine cells in the hepatic sinusoids in 19 species of teleosts but not directly in zebrafish. Because they were looking in very restricted areas of the liver for Kupffer cells, this could be the reason that they would have missed these macrophages. Currently, a formal and careful analysis of the development of Kupffer cells in zebrafish is lacking, but the first direct analysis of adult Kupffer cells was recently demonstrated (He et al., 2018), in agreement to our findings. To the best of our knowledge, the normal liver-resident macrophages characterization in Figure 1A represents the first dataset describing Kupffer cells in the developing zebrafish. To provide context to these results, we have added the following text in the Results section: “Prior to this work, Kupffer cells were thought to be missing or sparse in zebrafish and other teleost species (Goessling and Sadler, 2015; Sakano and Fujita, 1982), but a recent study tracing adult zebrafish Kupffer cells to their hematopoietic origin (He et al., 2018), and the data presented here collectively provide the first evidence for the prevalence of Kupffer cells in the zebrafish liver akin to their mammalian counterpart.”

6) Figure 2: an assessment of the fraction of intracranially-injected LPS that leaks to the periphery is necessary.

We have now done several experiments to directly trace LPS passage in real time from brain to periphery. Quantification of relative amounts drained out has been calculated based on real time tracking of the fluorescence. Please see new additional data provided in Figure 2C-D, Figure 2—figure supplement 3 and the complementary Video 7.

7) The brain injection causes a temporary hole in the BBB. Can one exclude that this is how LPS reaches the periphery?

We have conducted a more thorough analysis of the brain microinjection method and LPS movement. The original supplemental data (previous Figure 1—figure supplement 5) that analyzed the passage of dextran is not representative of the movement of LPS, so we are replacing it with multiple new data files: Figure 2A-B, Supplementary Figure 2—figure supplement 1, Figure 2—figure supplement 2, Figure 2—figure supplement 3 and Video 7. Due to the nature of the previous imaging setup that required manual image acquisition, we resorted to using a more blunt-ended needle for better visualization purposes than the fine-tipped needle we use for our routine brain microinjections (see Figure 2C-D, Figure 2—figure supplement 3 and Video 7). Therefore, the “hole” as referred does not represent our typical brain injections. Our new tracing analyses address LPS drainage directly, and the movie acquisition has significantly improved using an automated application at a very rapid speed of 1 Hz (60x faster than previous study) to reliably trace LPS for a continued 1 hour of imaging (Video 7).

8) Figure 4D: since total neutrophil numbers increase in irf8 morphants, it is not clear if this result is significant. This measurement should be somehow normalized to the total neutrophil count.

This is a valid concern. Although irf8 deficiency is known to cause a significant increase in baseline numbers of neutrophils, we did not observe an apparent increase in neutrophils surrounding the liver after LPS compared with controls (Figure 4E and data not shown), and furthermore, the shift in neutrophil behaviors after LPS towards more “macrophage-like”, taking on a more transient and long-term association with the liver stood out from those infiltrating in controls (Figure 4J). These traits suggest distinct changes in neutrophils that may not be explained merely by an increased baseline number. Nonetheless, we cannot entirely rule out the possibility that an increased baseline neutrophil number could contribute to the increased liver infiltration, so we have added this point in the Discussion section. We believe the agreement in the result from the two different methods (irf8 MO and clodronate) logically lend support to our interpretation of the data, and have added text to point out the potential caveat in the Discussion section.

9) Figure 4—figure supplement 3: a 27% reduction in neutrophil numbers is rather low. Csf3r ablation is also known to have some effect on macrophage populations (Liongue et al. Blood 11::2535, 209). A quantification on the impact of macrophages is required here.

Good point. We have conducted this control experiment by using in vivo imaging in double transgenic zebrafish carrying both macrophage and neutrophil reporters for control and csf3r MO conditions. We found again an average reduction in neutrophil number at ~30% in csf3r morphants compared with controls, but no significant change in macrophage number, suggesting no general effect on macrophage cell number due to csf3r MO (data added to Figure 4—figure supplement 3C). We also want to add that although the reduction of overall neutrophil number is modest on *average* ~30%, the maximum decrease in our cohort was as high as a 65% drop in number (Figure 4—figure supplement 3). Our assessment did not show macrophage reduction as that indicated by Liongue et al., but matched the csf3r zebrafish mutant which was reported to have normal macrophage numbers but ~ 50% reduction in neutrophils (Pazhakh et al., 2017). Liongue et al., however, examined at younger stages using only the csf1ra (fms) marker which may explain for the difference, but these results appear overturned by their own later publication showing that csf3r mutants have only neutrophil-specific phenotypes, and normal macrophages (Liongue and colleagues in Basheer et al., 2019) consistent with Pazhakh et al., 2017. Taken together, csf3r perturbation appears to target neutrophils specifically.

10) The tnnt2a MO experiment is not conclusive, since circulation was blocked from the onset, resulting, at 4dpf, in a smaller brain, a strong edema, and probably general inflammation. Drug- mediated inhibition of heartbeat, or physical blockade of bloodflow, would be more convincing.

We are not aware of a more effective method better suited for this experiment that could block circulation without substantial levels of side effects (including pharmacological chemicals we have read). We would welcome any specific suggestion on a proven method to block blood flow in zebrafish. To our knowledge, tnnt2a MO has been the most robust and frequently used for the purpose of stopping blood flow. It indeed affects development since blood flow is completely blocked from the beginning, but it does still allow us to perform an injection of LPS into the brain tectum to examine liver effects and macrophage behaviors, because these structures are still intact. For example, we found few to zero macrophages in the liver of tnnt2 morphants after brain-LPS injection similar to control brain-water injections despite presence of macrophages all around the liver (Figure 2—figure supplement 4). Despite the developmental hindrance due to a lack of blood flow, we think this reagent is effective for our intended purpose.

[Editors’ note: what follows is the authors’ response to the second round of review.]

Summary:Two new reviewers and one reviewer that evaluated the previous version of this work agreed in that the manuscript has been greatly improved with substantial new data and an in-depth revision. Two key experimental manipulations, the knock down of myd88 and il34, are now backed by stable loss of function mutations and several experiments have been strengthen with new or improved analyses. However, there are still two important experimental manipulations using morpholinos that have not been properly controlled. In addition, editorial changes are needed to better explain the use of LPS injections as an experimental tool.Essential revisions:1) The authors use morpholinos targeting csf3r and pu.1 expression and draw important conclusions based on those experiments. Given the inherent problems of morpholinos, particularly for inflammation studies, it is necessary to support the use of those reagents with stable mutants and/or additional controls. If mutants are not available or cannot be generated, the knockdown experiments may be further supported with rescue experiments and or F0 Criprs, in which case the significance of any findings related to those experiments should be tempered with an appropriate discussion of the caveats.

We have conducted new experiments to address reviewers’ concerns on the pu.1 and csf3r morpholinos.

1) We have obtained stable pu.1/spi1b mutants to support the pu.1 morpholino experiments. We used these mutants and their siblings as controls to examine the transcriptional impact of immune infiltration on liver genes, acute phase response, and inflammatory response, previously answered by the pu.1 morpholino manipulation. We present the pu.1 mutant data in Figure 5, and the pu.1 morpholino data previously in Figure 5 has been moved to Figure 5—figure supplement 1. Text has been added to discuss the data in the Results section. Overall, we found that data from the *pu.1* mutants corroborated and strengthened the effects we observed in *pu.1* morpholino-injected after brain-LPS injection.

2) Unfortunately, it has not been possible to obtain the stable csf3r mutant lines from either of the two currently published sources (Basheer et al., 2019; Pazhakh et al., 2017). We therefore have used F0 Crispr injection targeting the csf3r to address neutrophil effects on macrophage infiltration in the liver after brain-LPS injection to further support the csf3r morpholino experiments. We provide molecular analysis showing that the gene editing in csf3r was highly efficient in causing missense mutations and premature termination in the F0 Crispr injected animals followed by our phenotypic analysis. All data are presented in Figure 4—figure supplement 4.

We add the following text in the paper to describe the results as well as possible caveats due to the nature of the partial knockdown.

“As a complementary approach, we assessed the effect of *csf3r* gene disruption by Crispr injection on macrophage infiltration after brain-LPS injection and also found a significant decrease (Figure 4—figure supplement ). Both morpholino- and F0 Crispr injection- mediated *csf3r* knockdown corroborate to suggest a possible neutrophil role in recruitment of macrophages into the liver during inflammation. While neither method yields a total *csf3r* elimination to deplete most neutrophils, a stronger hindrance to macrophage infiltration may be possible from a complete *csf3r* depletion.” Additionally, we have edited text to soften the conclusion made from csf3r knockdown, instead of a requirement for neutrophil signaling for macrophage infiltration of liver, our revised text states macrophages coordinate with neutrophils. These changes were also made in the Abstract and the subsection “Liver infiltration by macrophages may be driven and coordinated by neutrophils”.

2) While brain injections of LPS can be a useful tool as used in this work, it is hardly a physiological condition. An editorial revision should address caveats and limitations, perhaps highlighting the use of this experimental approach as a tool.

Point well taken; thank you for the great suggestion. We have made several text modifications in the manuscript to address its limitations and highlight the use of LPS as our tool to trace molecules starting from the brain to understand its possible impact on the periphery. We list here all the major edits.

In the Abstract we have modified the text to say: “Using traceable lipopolysaccharides (LPS), we reveal that drainage of these inflammatory macromolecules from the brain led to a strikingly robust peripheral infiltration of macrophages into the liver independent of Kupffer cells.”

In the Introduction, we highlighted it as a tool: “To this end, we investigated if a brain perturbation triggering inflammation, using a brain-localized LPS microinjection as an experimental means, could trigger a peripheral organ response mediated by macrophages. [….] By using fluorescently traceable LPS in the brain as an experimental paradigm, we show that inflammatory cues could originate from the brain and trigger immune infiltration of the liver, a process not previously appreciated involving drainage of the macromolecules from brain to circulation.”

In the Discussion section, we provide a more thorough explanation as well as limitations to the use of LPS: “LPS were used as an effective tool for brain-localized microinjection in this study to allow tracing of inflammatory molecules originating in the brain to assess peripheral consequences, but whether endogenous proteins and other molecules could be secreted by the brain and drained into circulation to cause similar effects as the LPS do on the liver remains to be investigated. Some examples that lend support to this possibility relate to hallmarks of Alzheimer’s disease [...]”